**DOI: 10.1038/ncomms15738**　　**OPEN**

# Serotonin modulates a depression-like state in *Drosophila* responsive to lithium treatment

Ariane-Saskia Ries[1], Tim Hermanns[1], Burkhard Poeck[1] & Roland Strauss[1]

Major depressive disorder (MDD) affects millions of patients; however, the pathophysiology is poorly understood. Rodent models have been developed using chronic mild stress or unavoidable punishment (learned helplessness) to induce features of depression, like general inactivity and anhedonia. Here we report a three-day vibration-stress protocol for *Drosophila* that reduces voluntary behavioural activity. As in many MDD patients, lithium-chloride treatment can suppress this depression-like state in flies. The behavioural changes correlate with reduced serotonin (5-HT) release at the mushroom body (MB) and can be relieved by feeding the antidepressant 5-hydroxy-L-tryptophan or sucrose, which results in elevated 5-HT levels in the brain. This relief is mediated by 5-HT-1A receptors in the α-/β-lobes of the MB, whereas 5-HT-1B receptors in the γ-lobes control behavioural inactivity. The central role of serotonin in modulating stress responses in flies and mammals indicates evolutionary conserved pathways that can provide targets for treatment and strategies to induce resilience.

[1] Institut für Entwicklungsbiologie und Neurobiologie, Johannes Gutenberg-Universität Mainz, Colonel-Kleinmann-Weg 2, D-55099 Mainz, Germany. Correspondence and requests for materials should be addressed to R.S. (email: rstrauss@uni-mainz.de).

Animal models can help us to understand the neurobiological basis of mood disorders and aid in developing intervention strategies. Several stressors, genetic and pharmacological manipulations are used to induce a behavioural or motivational state that corresponds to criteria diagnosed in human patients (for example, anhedonia). Depression-like symptoms can be induced by chronic stress, maternal deprivation, learned helplessness, sleep deprivation, shifting light/dark cycles or olfactory bulbectomy[1,2]. One of the first animal models of major depressive disorder or MDD was developed by Seligman and colleagues by exposing dogs to repetitive and unavoidable electric shocks. These animals learned that this stressor cannot be controlled and later refrained from escaping/avoiding it even if given the chance to do so[3]. Following these initial experiments, the concept of learned helplessness has been extensively used to model MDD in rodents; shock-treated animals display similar pathophysiological (for example, dysfunction of 5-HT signalling) and behavioural changes (for example, general lack of motivation, sleep disruption, cognitive deficits) as observed in human patients[4]. Learned helplessness can also be induced in *Drosophila* flies by uncontrollable heat stress in a classical 'yoke' experiment, where a primary fly can, in a closed-loop, evade a heat-punishment by continuous walking (idle periods < 1 s); the same amount and time structure of heat is submitted to a second, yoked fly. However, as the yoked fly is unable to control the heat, it learns its own inability to control the punishment; walking activity declines during training (10 min) and stays low during the test minute thereafter[5]. Similar results were obtained when electroshocks were used as a punishment. Shortly after the induction of learned helplessness, yoked flies showed reduced walking activity and were deficient in place learning[6]. However, whether these protocols induce a general and lasting reduction in activity or lack of motivation to perform other behaviours has not yet been studied. Considering the evolutionary conservation of most of the neurotransmitter systems between vertebrates and flies, we set out to develop a paradigm, which might induce similar, long lasting behavioural deficits as observed in stressed mice.

In this current study we present a novel protocol for establishing a depression-like state in *Drosophila melanogaster* flies by repetitive episodes of uncontrollable 300 Hz vibrations over several days. We show that climbing decisions at a just insurmountable gap are a reliable readout for the motivational state of a fly. In the depression-like state flies show less climbing attempts than unstressed, sated control flies, whereas unstressed but hungry flies attempt to climb more often. The depression-like state reduces activity also in other voluntary behaviours, whereas it leaves more reactive behaviours like escape running unchanged. The stress-induced depression-like state is distinct from sexual deprivation following sexual rejection by mated females, which does not affect climbing decisions, and it can be reverted by feeding several different antidepressants and by feeding sugars, even without caloric value. By neurogenetic and pharmacological intervention we show that behavioural activity is regulated by serotonergic signalling to α- and γ-lobes of the mushroom bodies via different 5-HT receptors.

## Results

**Vibration stress induces a depression-like state in flies**. To establish a model for stress-induced mood disorders, we subjected male flies, grouped in small test tubes, to highly repetitive episodes of uncontrollable mechanical stress (300 Hz vibration) for ∼10 h a day (Supplementary Movie 1). To assess their motivational state, we monitored the number of climbing attempts at an insurmountable gap of 4.5 mm that just exceeds

their body reach. We chose this assay because climbing decisions can be reliably scored by the fly's typical leg-over-head movements when trying to climb a gap[7] (Fig. 1a inset). Moreover, the motivation of a fly to climb insurmountable gaps can be stimulated by pleasant odours and/or attractive landmarks on the opposite side of the cat walk (Supplementary Fig. 1). We suspected that the mushroom bodies (MB), a central brain structure involved in sleep homeostasis, temperature preference and olfactory memory formation[8,9] might be regulating the climbing motivation. Notably both olfactory and visual stimulation is mainly mediated through output signalling of the α-/β-lobes of the MB[9], because neuronal silencing of this compartment through the expression of tetanus toxin[10] in the MB suppressed the stimulatory effects of either cue (Supplementary Fig. 1b,c).

Over the course of 3 days of vibration-stress we observed a continuous reduction of climbing attempts from 50 to 30% in stressed flies (Fig. 1a). To assess if this chronic stress induces a general lack of motivation, we measured spontaneous 15 min walking activity in the Buridan paradigm[11,12] and the motivation (time until a males starts) to court a female[13]. Both innate behaviours were reduced in chronically stressed flies in comparison to unstressed controls (Fig. 1b,c). This was not due to fatigue or injury because when startled, these flies showed the same escape response in the fast-phototaxis assay as unstressed controls[14] (Fig. 1d). A test for optomotor response[15] supported the notion that only voluntary behaviours are affected by vibrational stress, because stressed flies compensated the rotatory visual stimulation to the same extent as controls (Fig. 1e).

A typical symptom of MDD is the lack of interest in enjoyments, (anhedonia[1]), like the loss of appetite. To test for a similar response in stress flies we developed the so-called stop-for-sweet paradigm. Hungry flies performing negative geotaxis in an upright microtiter plate were given the opportunity to feed on a stripe of sweet tasting glycerol. In contrast to controls stressed flies stopped less frequently, indicating that they have lost their interest in sweets (Fig. 1f). In summary, enduring, uncontrollable stress over several days can induce a depression-like state in flies that reduces voluntary, innate behaviours. Notably, this is not due to fatigue because behaviours induced by exogenous stimuli, like optomotor compensation and startle response in the fast phototaxis, are not affected.

In addition, the lack of motivation to climb after chronic stress is not a general response to any negative experience, because males who experienced 4 days of sexual deprivation and defeat by unreceptive females[16] showed unchanged motivation for gap-climbing (Fig. 1g). Moreover, in contrast to the males that were sexually rejected and deprived, overexpression or knockdown of neuropeptide F[16] did not alter the vibration-induced depression-like state (Fig. 2a,b). Notably, in contrast to helplessness or conditioned courtship suppression[17], this state does not seem to have a learning component because learning mutants, deficient in cAMP-synthesis (*rutabaga[1]*) or -metabolism (*dunce[1]*)[9], were similarly susceptible to stress like wild-type flies (Fig. 2c,d). So far, long-term memory formation in *Drosophila* has been shown to require cAMP signalling which results in the activation of the transcriptional regulator CREB[9] suggesting, that either an alternative biochemical pathway consolidates learned elements of the depressive state, or stress-induced lack of motivation is not a learned behaviour in flies.

Lastly, we tested if the motivation deficit is amenable to pharmacological treatment as shown for MDD patients[18]. Therefore, we tested if the mood stabilizing salt lithium

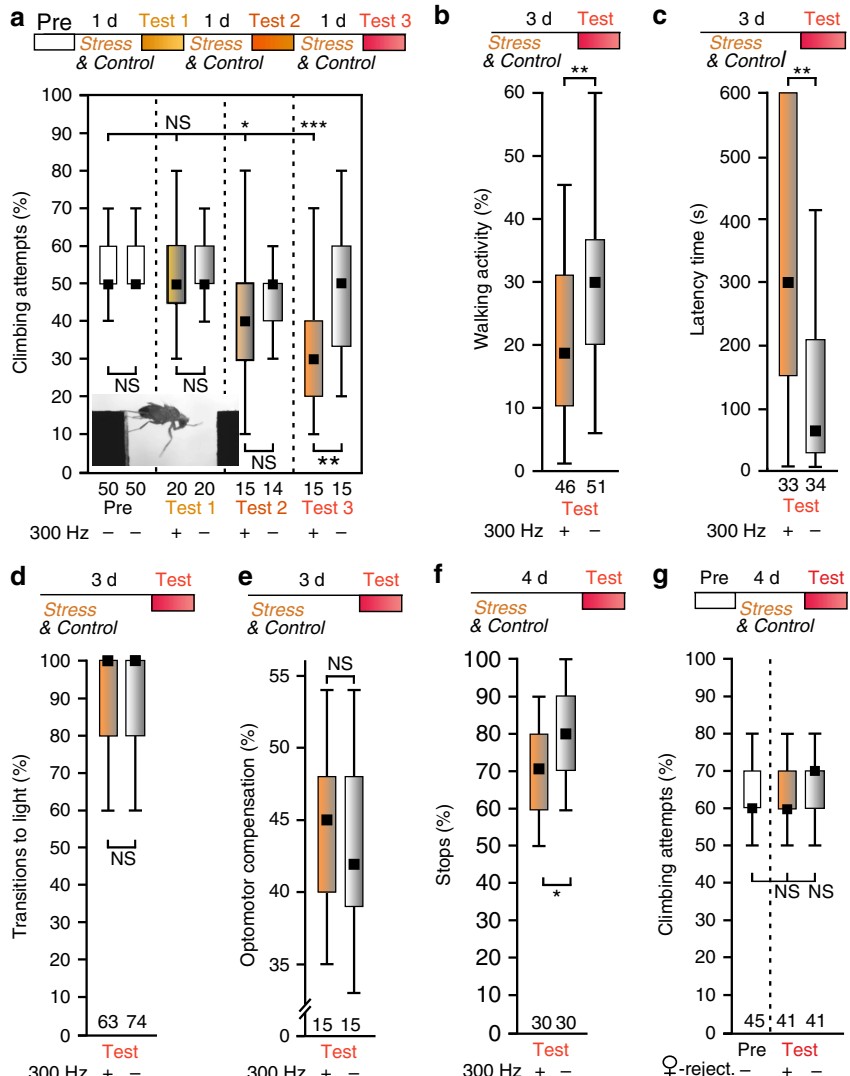

**Figure 1 | Repeated inescapable vibrations induce depression-like state in wild-type CS flies.** (**a**) Males were tested for their motivation to climb (Pre-test) an insurmountable 4.5 mm-wide gap in a catwalk (inset). Decreasing fractions of climbing attempts were observed after, 2 days (Test 2), or 3 days (Test 3) of vibration treatment (orange bars), but not in sham-treated males (grey bars). (**b**) Fraction of time spent walking within 15 min in Buridan's paradigm is lower in vibration-stressed flies. (**c**) Latency time to initiate the first courtship song is longer in vibration-treated males than in controls. (**d**) Normal running ability in fast phototaxis shows that vibration treatment does not physically harm the flies or reduce their escape response. (**e**) Stressed flies are able to compensate a rotatory visual stimulus as good as controls. (**f**) Hungry flies performing negative geotaxis stop at stripe soaked with glycerol, whereas vibration-treated males tend to ignore the sweet taste and continue their walk. (**g**) Eight courtship-suppression sessions with unreceptive females do not induce a depression-like state in gap-climbing males. Black squares, medians; boxes, 25 and 75% quartiles; whiskers, 10 and 90% quantiles; numbers of animals are indicated below each box. NS, not significant; *, $P<0.05$; **, $P<0.01$; ***, $P<0.001$; Mann–Whitney $U$ tests.

chloride (LiCl), which is used as an augmenting medication for MDD in humans, can also change the depression-like state in flies. After 3 days of stress application, we fed 50 mM LiCl (a dose effective in flies[19,20]) overnight and retested the flies (after 4 h of stress) the next day. Indeed, LiCl-treated flies were relieved from the depression-like state; however, stressed and control flies showed excessive climbing behaviour (Fig. 3a). Feeding just 5 mM LiCl did not induce this manic behaviour in the control flies but still effectively relieved stressed flies, which showed the same climbing motivation as pre-test controls (Fig. 3b). Both the induced manic behaviour at high doses and the amelioration of the stress-induced phenotype suggests that evolutionary conserved biochemical pathways are involved in MDD and in the depressive-state of flies opening up the possibility to study genetic underpinnings of Li toxicity[21] and the mode of function in MDD therapy[22].

**Uncontrollable stress induces reduced 5-HT signalling.** Accumulating evidence points to a central role of the neurotransmitter serotonin (5-HT) in MDD aetiology in mammals[23]. Reduced levels of 5-HT or its precursor L-tryptophan have been described in MDD patients and dietary-induced depletion of L-tryptophan can induce similar symptoms as observed in depression[23]. Although this simplistic view on the role of 5-HT is controversial, changes in 5-HT signalling and 5-HT-1A receptor function have been well established in animal models of MDD[24]. Therefore, we asked if reduced 5-HT signalling might be involved in our fly model and fed stressed flies with the 5-HT-precursor 5-hydroxy-L-tryptophan (1 mg ml$^{-1}$ 5HTP) in sucrose solution overnight. On the next day, after 4 h of stress application, the same flies were retested and showed relief from the depression-like state, but notably the same result was achieved with the 5%-sucrose solution alone (Fig. 4a).

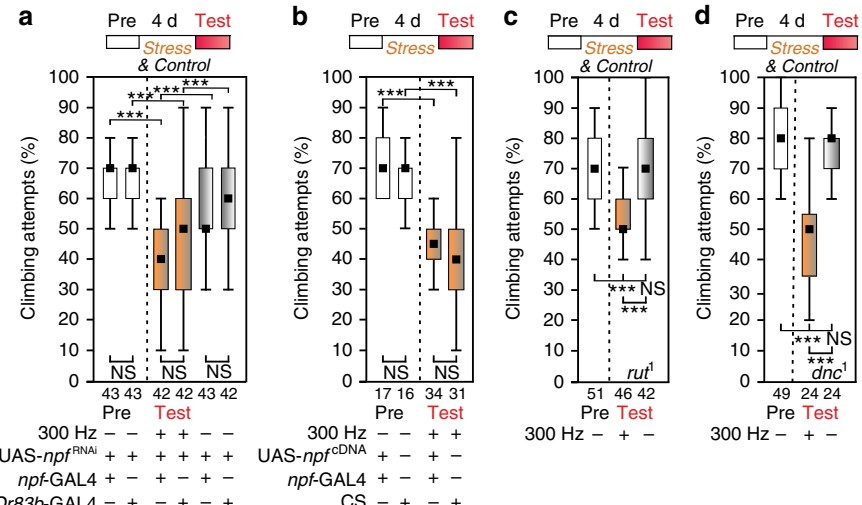

**Figure 2 | Depression-like state establishes independent of the Neuropeptide-F (NPF) and the cAMP-pathway. (a)** Knockdown of *npf* in its endogenous pattern by *npf*-GAL4 (ref. 50) driving UAS-*npf* [RNAi] can neither prevent nor reinforce the depression-like state whereas sham-treated flies stay active. As a negative control UAS-*npf* [RNAi] has been driven in olfactory receptor neurons via Or83b-GAL4. **(b)** Overexpression of NPF by driving *npf*-cDNA in its endogenous pattern via *npf*-GAL4>UAS-*npf* [cDNA] cannot either prevent a depression-like state after 4 days of vibration treatment. Wild-type CS for comparison. **(c,d)** Learning mutants that cannot be conditioned by sexual rejection and deprivation *rut*[1] (*rutabaga* encodes for an adenylyl cyclase) and *dnc*[1] (*dunce* encodes for a phosphodiesterase) develop the depression-like state after 4 days of vibration treatment. Sham-treatment for control. Black squares, medians; boxes, 25 and 75% quartiles; whiskers, 10 and 90% quantiles; numbers of animals are indicated below each box; NS, not significant; ***, $P < 0.001$; Kruskal–Wallis test.

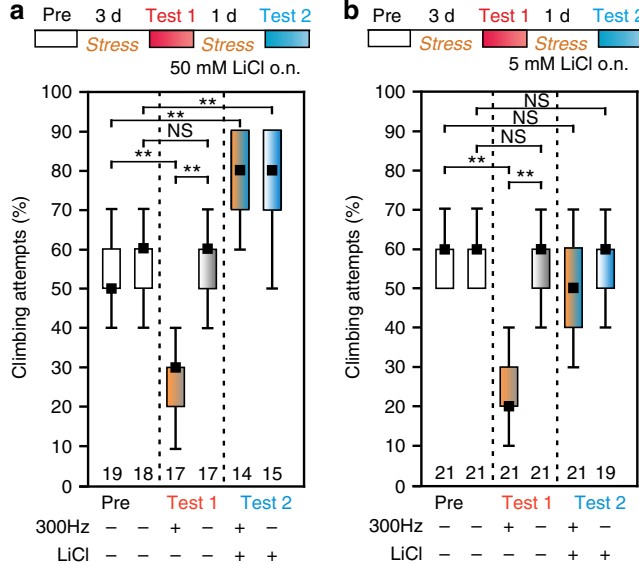

**Figure 3 | Lithium chloride ameliorates the depressive-like state.** **(a)** Overnight treatment with a 50 mM LiCl water solution induces excessive climbing attempts at the 4.5 mm gap in stressed and control flies. **(b)** At the lower concentration of 5 mM control flies show the same motivation to climb (Test 2) as in the pre-test. Vibration-treated flies recover from the depressive-like state (Test 1 versus Test 2) and show a similar motivation to climb as in the pre-test. Black squares, medians; boxes, 25 and 75% quartiles; whiskers, 10 and 90% quantiles; numbers of animals are indicated below each box; NS, not significant; **, $P < 0.01$; Kruskal–Wallis test.

Moreover, supplying stressed flies with a 30 min-supplement of sucrose after each daytime stress treatment prevented the depressive-like state; that is, conveyed resilience to continuous stress (Supplementary Fig. 2a). To exclude that this sucrose relief is based on the additional caloric intake, we repeated the experiment and fed 5% arabinose, a sugar that is similarly sweet tasting but has no nutritional value for the fly[25]. Arabinose conveyed a comparable increase in climbing activity to stressed flies like sucrose (Supplementary Fig. 2b), supporting the idea that the 'pleasure' of sweet sensation can relieve flies from the depression-like state.

To determine whether the relief by sucrose is mediated via 5-HT signalling, we added 50 mM of the 5-HT synthesis inhibitor α-methyl-DL-tryptophan (α-MTP) to the sucrose solution, which abolished the stress relief (Fig. 4b). To verify the 5-HT-inducing effect of sucrose, we performed anti-5-HT labelling in the adult brains and compared it to 189Y-GAL4 induced mCD8::GFP in the α-/β-lobes of the MBs and in ring neurons of the ellipsoid body (EB)[26]. Indeed, as reported earlier by Chiang and colleagues (in their supporting information)[27], feeding 5% sucrose overnight elevated 5-HT signals in the brain and adding the α-MTP inhibitor to the sugar solution impeded this effect (Fig. 4c; quantification in Supplementary Fig. 2c,d). To assess whether vibration-stress reduces 5-HT levels, we compared anti-5-HT immunoreactivity with mCD8::GFP levels in the EB of stressed and unstressed flies (Fig. 4d,e) but did not find a significant difference (GFP expression was also not altered in stressed flies; see Supplementary Fig. 2e). However, comparing 5-HT levels in the α-/β-/γ-lobes of the MBs with those in the EB in stressed flies revealed a 21.3% reduction at the α-lobes, but no significant changes at the β-/γ-lobes (Fig. 4f–h), suggesting that 5-HT release at the α-lobes was compromised during continuous stress application.

The whole MB is innervated by a subset of serotonergic neurons and blocking the chemical output-synapses of these neurons by expressing tetanus toxin[10] with *Trh493*-GAL4 (ref. 27) did not induce an instantaneous depressive state. However, these flies were unreceptive to the ameliorating effect of sucrose (Fig. 5a), suggesting that activity-dependent release of 5-HT from these neurons is mediating the sugar relief. Supporting this hypothesis, tonic depolarization of the *Trh493*-neurons by expressing the bacterial sodium-ion channel

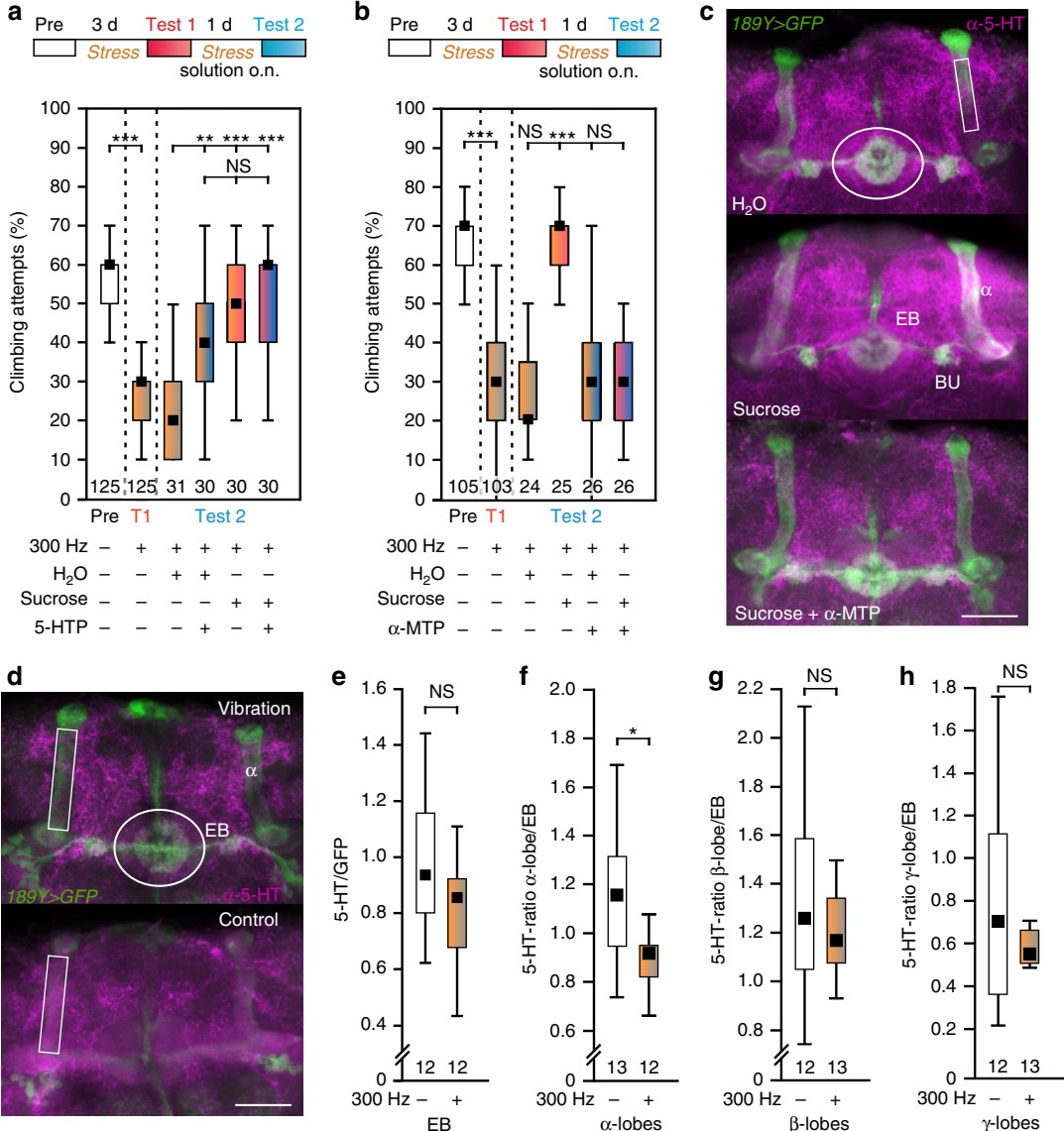

**Figure 4 | Serotonin level in mushroom body α-lobes correlates with the depression-like state.** (**a**) Reduced fraction of climbing attempts after 3 days of vibration treatment is ameliorated by feeding the serotonin precursor 5-hydroxy-L-tryptophan (5-HTP; 1 mg ml$^{-1}$) overnight. Sucrose solution (5%) or sucrose solution with 5-HTP overnight ameliorate insignificantly more effectively (Kruskal–Wallis test). (**b**) Sucrose solution ameliorates the depression-like state; however, not when the 5-HT-synthesis blocker α-methyl-DL-tryptophan (α-MTP; 20 mM) is added (Kruskal–Wallis test). (**c**) Sucrose treatment is signalled via 5-HT release in the fly brain. Three 22.5 μm thick layers of confocal stacks stained with 5-HT antibody (magenta) showing the ellipsoid body (EB; ring), R3 ring neurons (dendrites in the bulbs, BU), and α-lobes of mushroom bodies merged with the expression pattern of 189Y-GAL4 > UAS-mCD8::GFP. Fly brains were prepared on day 4 after 3 days on food and the following night either on water, on 5% sucrose solution, or on 5% sucrose solution with 20 mM α-MTP; quantification of 5-HT labelling divided by the GFP signal at the indicated regions in Supplementary Fig. 2. (**d–h**) 5-HT in flies in a depression-like state compared to controls. Elliptic region of interest around EB is compared to rectangular regions around each mushroom-body lobe (example in **d** shows α-lobes). Grey values of all pixels with common x-/y-coordinates were summed up along the z axis (creating voxels) and voxel means determined. (**e**) Relative 5-HT content of EB; GFP-label for reference. (**f–h**) '5-HT lobe staining' in the indicated mushroom body lobes divided by '5-HT EB staining'. Black squares, medians; boxes, 25 and 75% quartiles; whiskers, 10 and 90% quantiles; numbers of animals are indicated below each box; NS, not significant; *, $P < 0.05$; **, $P < 0.01$; ***, $P < 0.001$; Mann–Whitney U test was used in **e–h**. Scale bars denote 25 μm.

NaChBac[28] prevented the depression-like state and feeding sucrose did not further increase the resilience against stress (Fig. 5b). As expected, when 5-HT synthesis was blocked after the vibration treatment by adding α-MTP to the sucrose solution, Trh493 > NaChBac flies were again susceptible to the vibration stress.

**Differential modulation by mushroom-body compartments.** Because the motivation to climb is modulated by the α-/β-lobes

of the MB (Supplementary Fig. 1), we asked which parts of the MB require serotonergic signalling to convey sugar-mediated resilience to vibration stress. Trh493-neurons preferentially contact neurons at the α- and γ-lobes[27] but not at β-lobes (Fig. 5c–e), as shown by co-staining of Trh493 > mCD8::GFP, and anti-Fascilin II (α-/β-lobes) or anti-Trio (γ-lobes)[29]. Blocking the chemical synapses of the target cells of Trh493 neurons in the α-/β- and γ-lobes (using mb247-GAL4 (ref. 30) prevented the depression-like state (Fig. 5f). Distinct α- and γ-lobe functions became evident when we silenced only the γ-lobe output

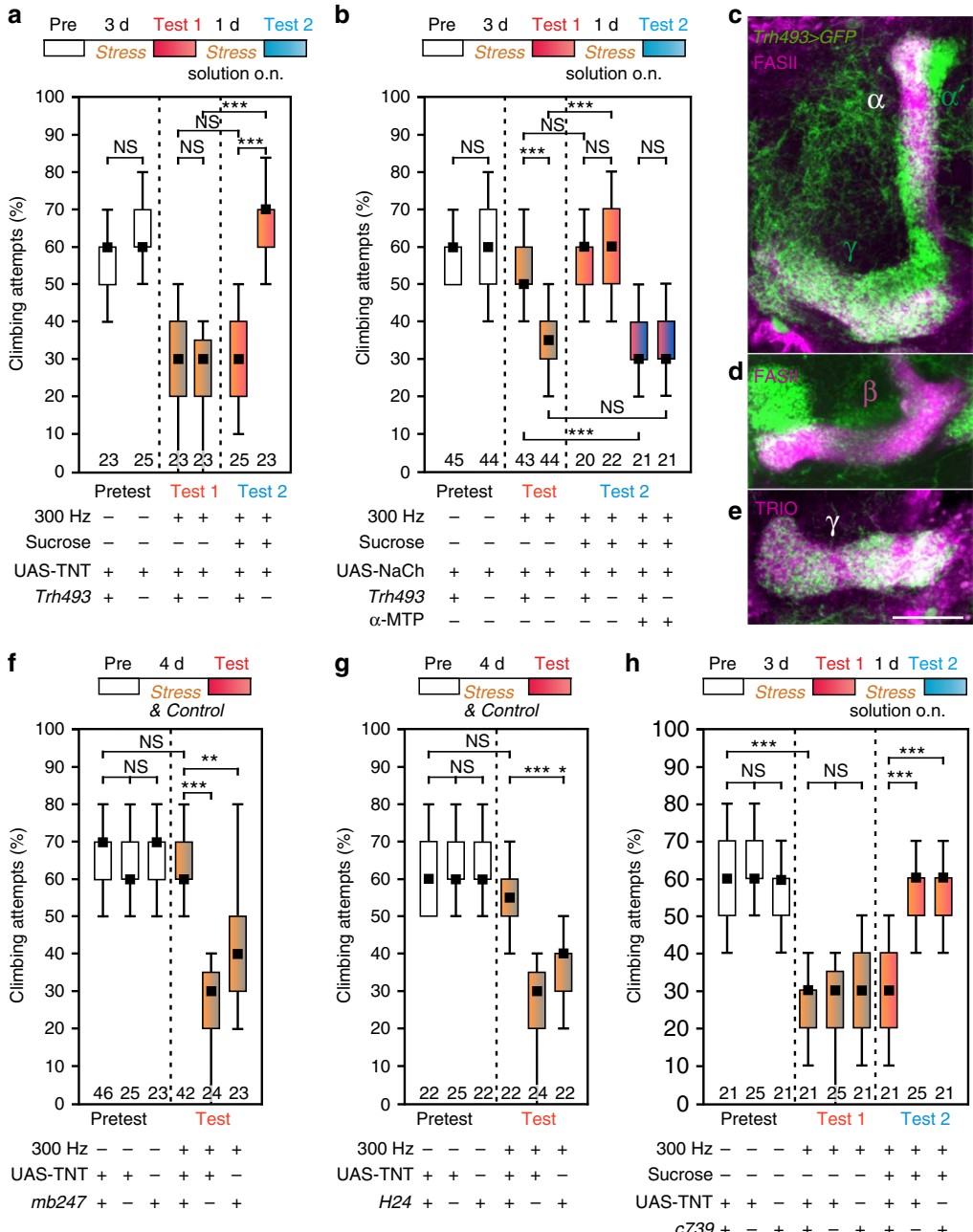

**Figure 5 | Serotonergic neurons signal onto mushroom body α- and γ-lobes to modulate the depression-like state.** (**a**) Blocking chemical synapses of serotonergic *Trh493*-GAL4 neurons by expressing TNT prevents the amelioration of the depression-like state by sucrose treatment overnight from day 3 (Test 1) to day 4 (Test 2), whereas driverless UAS-TNT controls recover by sucrose solution overnight. (**b**) Tonic depolarization of serotonergic *Trh493*-GAL4 neurons by expressing bacterial sodium-ion channels UAS-NaChBac prevents the depression-like state after 3 days vibration treatment; controls lack the driver. Sucrose overnight does not increase the effect of depolarizing *Trh493* neurons. After blocking of 5-HT synthesis (α-MTP overnight), neither depolarization of *Trh493* neurons nor sucrose treatment can ameliorate the depression-like state. (**c–e**) Confocal stacks of 10.5 μm thickness, (**c**) immediately rostral to **d**. Anti-Fasciclin II staining (FASII, magenta) shows α-/β-lobes and (**e**) Anti-TRIO labelling the γ-lobes. *Trh493*-GAL4 driving UAS-mCD8::GFP shows co-localization with α- and γ-lobes, but not with β-lobes. The scale bar denotes 25 μm. (**f**) Blocking chemical synapses of the mushroom body Kenyon-cells in the *mb247*-GAL4 pattern (α-/β-/γ-lobes) prohibits the depression-like state even after 4 days of vibration treatment. Controls lack driver or effector. (**g**) Corresponding experiment to **f** but with γ-lobe Kenyon-cell driver *H24*-GAL4; γ-lobes are relevant for bringing about the depression-like state. (**h**) Blocking of α-/β-lobe Kenyon-cells in the pattern of *c739*-GAL4 cannot prevent the depression-like state after 3 days of vibration treatment; however, amelioration by sucrose overnight is abolished. Controls lack driver or effector. Black squares, medians; boxes, 25 and 75% quartiles; whiskers, 10 and 90% quantiles; numbers of animals are indicated below each box; NS, not significant; *, $P < 0.05$; **, $P < 0.01$; ***, $P < 0.001$; Kruskal–Wallis test.

(*H24*-GAL4 > UAS-TNT (ref. 30)), which conveyed resilience to the vibration stress (Fig. 5g). In contrast, blocking the output of the α-/β-lobes (*c739*-GAL4 > UAS-TNT (ref. 30)) rendered the flies susceptible to stress but unresponsive to the sucrose relief (Fig. 5h). Note that silencing of either compartment did not significantly change the endogenous propensity to climb the gap. This suggests that the MB does not just trigger this behaviour upon a visual cue, but rather modulates the motivation to climb.

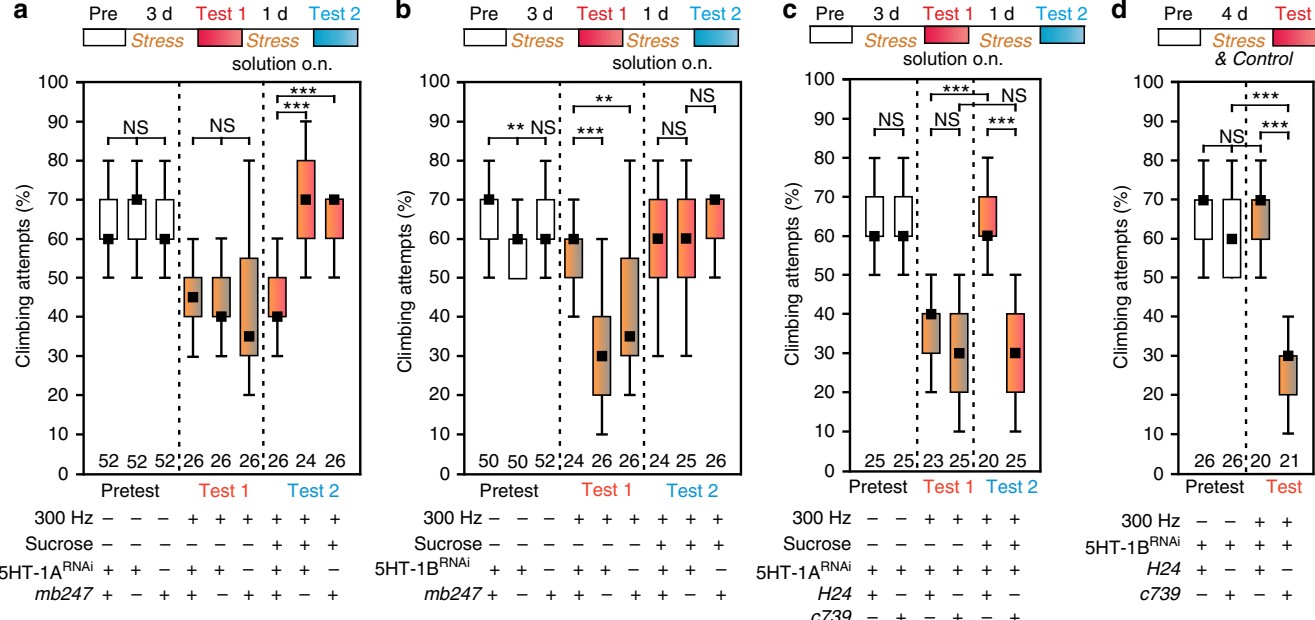

**Figure 6 | Relevance of 5-HT-1A receptors in α-lobes and 5-HT-1B receptors in γ-lobes of the mushroom body.** (**a**) Knockdown of 5-HT-1A receptors via RNA-interference (UAS-5HT-1A^RNAi, #25834) in *mb247*-GAL4 mushroom body α-/β- and γ-Kenyon-cells prevents sucrose relief from depression-like state. (**b**) Knockdown of 5-HT-1B receptors (UAS-5HT-1B^RNAi, #27634) in *mb247*-GAL4 Kenyon-cells prohibits stress-induced depression. Effector and driver controls show the depression-like state which is reversible by sucrose treatment. (**c**) Specific knockdown of 5-HT-1A receptor (UAS-5HT-1A^RNAi, #33885) expression in the α-/β-lobes (*c739*-GAL4), but not in the γ-lobes (*H24*-GAL), prevents the sucrose relief after vibrational stress. (**d**) Reducing 5-HT-1B expression (UAS-5HT-1B^RNAi, #54006) in the γ-Kenyon cells renders the flies resilient to the vibration stress. Knockdown of the 1B-receptor in the α-/β-Kenyon cells is inconsequential. Black squares, medians; boxes, 25 and 75% quartiles; whiskers, 10 and 90% quantiles; numbers of animals are indicated below each box; NS, not significant; **, $P < 0.01$; ***, $P < 0.001$; Kruskal–Wallis test.

To further address distinct roles of MB lobes we focused on two serotonin-receptor types, 5-HT-1A and 5-HT-1B that are expressed in α/β- or γ-Kenyon cells of the MB, respectively[19,31–33] (see Supplementary Fig. 3). Both *Drosophila* receptors inhibit adenylyl cyclase and, in contrast to vertebrates, activate phospholipase C[34]. Via this IP_3-pathway, 5-HT-1A- and 5-HT-1B-receptor activation leads to strong intracellular calcium signals[33]. A knockdown of 5-HT-1A via RNA-interference (RNAi)[35–37] in α-/β- and γ-lobes (*mb247*-GAL4) did not change the susceptibility to stress but prevented the sugar relief (Fig. 6a). In contrast, knockdown of 5-HT-1B receptors[19] via RNA-interference in *mb247*-GAL4 Kenyon-cells (Fig. 6b) prevented the stress-induced depressive-like state, suggesting that the MB compartments exert a push–pull modulation of climbing motivation through serotonergic activation of the α-/β- and γ-lobes, respectively.

To ensure that these phenotypes are not due to developmental defects or to continuous malfunction of adult Kenyon cells, we repeated the experiment and confined the RNAi mediated knockdown to the adult state employing a temperature-sensitive repressor transgene *Tub* > GAL80^ts (ref. 38 and obtained similar results (Supplementary Fig. 4). Moreover, to restrict the knockdown of both 5-HT receptors to specific compartments we induced RNAi in the α/β-cells (*c739*-GAL4) and γ-cells (*H24*-GAL4) of the MB, respectively. As expected, reduced expression of 5-HT-1A receptor in the α/β-Kenyon cells prevented the sugar relief, whereas knockdown of 5-HT-1B rendered the flies resilient to the vibration stress (Fig. 6c,d).

As mentioned above, Kenyon cells with 5-HT-1A receptors can be addressed by *5-HT-1A*-GAL4 (refs 32,33) (Supplementary Fig. 3a) and tonic depolarization by expressing NaChBac induced resilience to stress treatment (Fig. 7a). Conversely, tonic depolarization of Kenyon cells expressing 5-HT-1B receptors via

*5-HT-1B*-GAL4 (refs 19,32) (Supplementary Fig. 3b) increased the susceptibility of flies for vibrational stress, because a single day of vibration effectively induced the depression-like state (Fig. 7b).

**Pharmacological manipulation of 5-HT signalling.** To further support our push–pull model of serotonergic signalling at the MB we performed two pharmacological experiments. Selective serotonin reuptake inhibitors (SSRI) like Fluoxetine reduce the reuptake of 5-HT from the synaptic cleft, thus increasing serotonergic signalling, and are widely used to treat MDD[23]. According to our hypothesis that reduced 5-HT signalling from the *Trh493*-positive neurons to the α-lobes of the MB is causal for the depression-like state, feeding 10 mM Fluoxetine to stressed flies should ameliorate the behavioural inactivity (a similar dose has been shown to affect light entrainment of the circadian clock in flies[19]). As shown in Fig. 7c, treatment with the SSRI significantly increased the climbing activity of stressed flies, yet to a lesser extent as 5% sucrose (Fig. 4b) or 5 mM LiCl (Fig. 3b). Nevertheless, this result supports the idea that increasing duration and/or amount of 5-HT signalling (at the α-lobes of the MB) is relieving from the depressive-state. Conversely, pharmacological activation of the 5-HT-1B receptor in the γ-cells of the MB should reduce climbing activity. The agonist 8-OH-DPAT has been shown to activate the 5-HT-1B receptor in flies[19] and, in contrast to vertebrates, has a low affinity to the 5-HT-1A receptor[34,39]. Feeding 3 mM 8-OH-DPAT overnight to control flies significantly reduced the climbing activity, whereas flies expressing the neuronal silencer TNT specifically in the γ-neurons (*H24* > TNT) were unaffected (Fig. 7d). Note that just silencing the γ-neurons of the MB did not change the climbing behaviour, but it prevented the negative serotonergic modulation of the 5-HT-1B agonist.

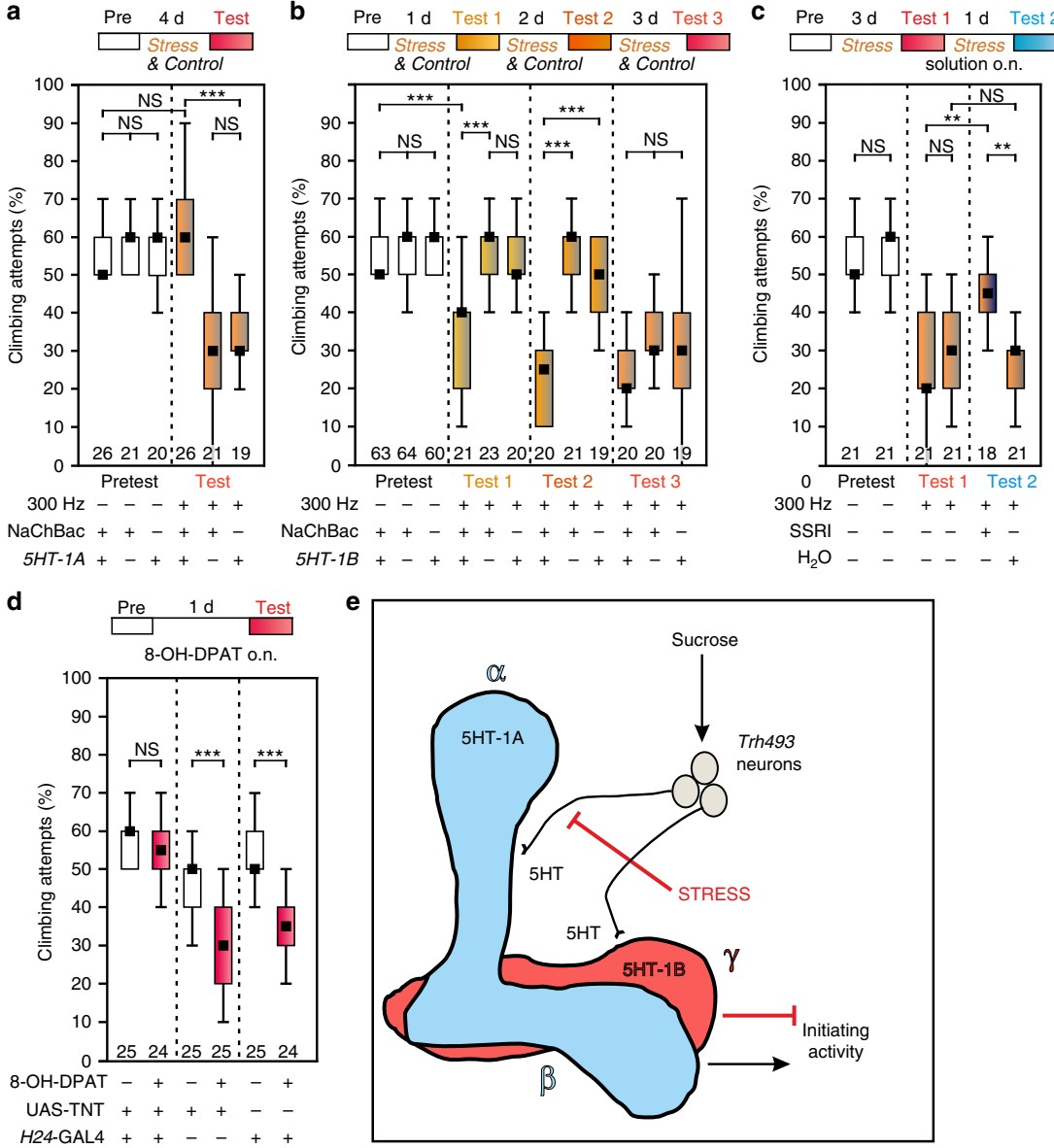

**Figure 7 | Genetic and pharmacological manipulations of 5-HT signalling reveals a dual role of the mushroom body.** (**a**) Tonic depolarization of 5-HT-1A receptor-expressing cells (*5HT-1A*-GAL4 > NaChBac) prohibits the depression-like state after vibration treatment. Effector and driver controls respond to stress treatment. (**b**) Tonic depolarization of 5-HT-1B receptor-expressing cells (*5-HT-1B*-GAL4 > NaChBac) increases susceptibility for vibrational stress in comparison to effector and driver controls; just 1 day instead of 3 days of vibration treatment is effective. (**c**) Treating stressed flies with the selective serotonin reuptake inhibitor (SSRI) Fluoxetine (10 mM overnight) ameliorates the depressive-like state. (**d**) Feeding the serotonin receptor agonist 8-OH-DPAT (3 mM overnight) reduces the climbing motivation of UAS-TNT and *H24*-GAL4 control flies. Neuronal silencing of the γ-neurons of the mushroom body (*H24*-GAL4 > TNT) eliminated this pharmacological action. Numbers of animals are indicated below each box; NS, not significant; **, $P < 0.01$; ***, $P < 0.001$; Kruskal–Wallis test. (**e**) Summary in a model. Sucrose reward is signalled to α- and γ-lobes of the mushroom body via 5-HT release from *Trh493* neurons. Mushroom body Kenyon cells possessing 5-HT-1A (α-lobes) or 5-HT-1B receptors (γ-lobes) receive this information. Activation of 5-HT-1A-expressing Kenyon cells enhances, activation of 5-HT-1B-expressing neurons inhibits initiating behavioural activity. Therefore the motivation to perform climbing behaviour is not changed if sucrose is fed to unstressed flies. Repeated vibrational stress inhibits 5-HT release to α-lobes, they become depleted of serotonin and can no longer enhance behavioural activity. This depression-like state is ameliorated by sucrose treatment because overall 5-HT levels are elevated thus rescuing the loss of 5-HT at the α-lobes.

## Discussion

In summary, serotonergic signalling to the MB has to be tightly balanced to generate adequate behavioural activity (Fig. 7e). Activation of α-lobe Kenyon cells expressing 5-HT-1A enhances behavioural and particularly climbing activity, whereas activation of γ-lobe 5-HT-1B-expressing neurons inhibits climbing activity. Sucrose treatment is signalled to most parts of the brain via serotonergic neurons, including the α-lobes via

*Trh493*-neurons. Since α-lobes (push) and γ-lobes (pull) are both receiving the signal, the behavioural activity remains unchanged. Repeated vibrational stress inhibits 5-HT release to MB α-lobes so that they become depleted of serotonin and can no longer enhance behavioural activity. Thus, rewarding and unpleasant influences are integrated via serotonin levels in the α-lobes. Either a knockdown of 5-HT-1B receptors or blocking the output of γ-lobe Kenyon cells expressing these receptors

turns flies resilient to continuous stress. Importantly, neither silencing nor tonic activation of the respective Kenyon cells changed the spontaneous rate of climbing attempts. This suggests that the MB does not elicit or prevent climbing, but rather modulates motivation to do so in response to environmental factors; that is stress, hunger or attractive targets. Similarly, dopaminergic signalling to the MB conveys environmental influences and the inner behavioural state of the fly, to modulate the behavioural response to odours[40]. This supports the interpretation that the insect MB corresponds to the vertebrate hippocampus and amygdala[41] and furthermore, that serotonin plays a similar role in stress response in flies as in vertebrates. Acute stress in mice and rats enhances serotonin levels in the hippocampus whereas chronic unpredictable stress induces depressive-like behaviour and reduces 5-HT release to the hippocampus and other brain regions[24]. The human 5-HT$_{1A}$ Gi/Go-protein-coupled receptor acts as autoreceptor in the dorsal raphe and has been associated with mood disorders and antidepressant response, because activity of serotonergic dorsal raphe neurons is negatively regulated by this receptor type upon stress[24]. In the hippocampus and medial prefrontal cortex region postsynaptic 5-HT$_{1A}$ receptors subsequently receive less serotonin after chronic unpredictable stress[24]. In *Drosophila* 5-HT-1A seems to fulfil a similar role as a postsynaptic receptor in the MB. The interpretation is supported by the pharmacological rescue of the depressive-like state by feeding the SSRI Fluoxetine to stressed flies. Reducing 5-HT reuptake from the synaptic cleft between serotonergic *Trh493*-neurons and the postsynaptic α/β-Kenyon cells of the MB should compensate for the reduced 5-HT release by the *Trh493*-neurons during permanent stress. The 5-HT-1B function in the γ-lobes might be analogous to the rodent 5-HT$_{1B}$ receptor because knockout mice show increased exploratory activity[42]. This suggests that the fly model can be used to study common signalling pathways underlying stress-induced 'depression', resilience to stress and control of motivation. Future studies will have to show whether additional neurotransmitter systems like the dopaminergic system are affected in the depression-like state of flies.

This conjecture is supported by the successful pharmacological suppression of the depressive-like state in flies by feeding the mood stabilizer LiCl. Lithium therapy of MDD has been applied for more than 50 years, but its complex action in the vertebrate brain is far from understood. Several neurotransmitter systems have been implicated to be affected by Li action, including serotonin[43]. Interestingly, Li ions have been shown to specifically inhibit the 5-HT$_{1B}$ receptor in a non-competitive fashion[44] and this observation from vertebrates would fit our push–pull model of climbing motivation in *Drosophila*. Reducing 5-HT-1B signalling in the γ-lobes of the MB could counteract the reduced serotonergic activation of α-lobes in stressed flies. However, feeding 10 or 50 mM LiCl (1day and 4 days, respectively) can also change the transcriptome in *Drosophila* heads[20,45], which could equally explain changes in behaviour. An established target of Li inhibition in *Drosophila* is the GSK3-β protein kinase Shaggy (SGG). Low doses of LiCl can prolong the lifespan of flies through SGG inactivation, whereas higher concentrations are contra productive[46]. Li application also seems to affect SGG function downstream of the 5-HT-1B receptor in clock neurons, expressing the neuropeptide PDF, that synchronize the circadian clock in the *Drosophila* brain[19]. Whether SGG is also a relevant downstream target of 5-HT signalling in the MB remains to be seen, however it is known that Li treatment modulates behaviour in rodents and there is increasing evidence for a role of the GSK-3/Wnt signalling pathway in mice models of bipolar disorders[47].

## Methods

**Fly stocks.** *Drosophila melanogaster* flies were raised on standard medium (water, cornmeal, soy bean, agar, molasses, yeast, preservative methyl-4-hydroxybenzoate) at 25 °C, 60% humidity and 14 h/10 h light–dark cycles. *Canton-S* (*CS*) served as wild-type *Drosophila* strain in this study. The following colleagues kindly provided additional stocks: D. Armstrong (*189Y*-GAL4), M. Heisenberg (wild-type Canton-S, *rut$^1$*, and *dnc$^1$*), T.J. Hirsh (*Trh493*-GAL4), Y. Rao (*5HT-1A*-GAL4), H. Scholz (UAS-TNT), P. Shen (UAS-*npf*), H. Tanimoto (*H24*-GAL4, *NP1131*-GAL4 and *mb247*-GAL4; *Tub*-GAL80$^{ts}$), and T. Zars (*mb247*-GAL4). The following lines were obtained from the Bloomington Stock Center: Or83b-GAL4 (#2681), c305a-GAL4 (#30829), c739-GAL4 (#7362), npf-GAL4 (#25681), 5HT-1B-GAL4 (#27637), UAS-NaChBac (#9469), UAS-5HT-1A$^{RNAi}$ (#25834 and #33885), UAS-5HT-1B$^{RNAi}$ (#27634 and #54006), UAS-CD4::tdTomato (#35837) and UAS-mCD8::GFP (#5130).

**Stress treatment.** Groups of 15–25 male flies of 3-to-5-days of age were placed in acrylic tubes of 98 mm length and 4 mm width. Cotton wool stoppers on both sides restricted the flies to a length of 25 − 30 mm. The acrylic tubes were either placed on a vibration device (Brüel & Kjoer, type 4810; stressed group) or put on a table surface (control group) for 8 h a day and 3–4 days during a week. Stressed and control flies always came from the same culture vials; assignment to a treatment was random. Vibrations of 300 Hz were applied for 20 s followed by a pause of 10 s to prevent habituation. The 20 s/10 s schema has been repeated for 15 min followed by a break of 30 min recovery time. Twelve to 14 cycles were applied per day. The control group was deprived of water and food during the same time spans. After 8 h in the acrylic tubes, animals of both groups were put back on standard cornmeal medium in culture vials overnight. Tests were done alternatingly. In the climbing assay the group affiliation became quickly obvious. For all other tests the technical helpers were blind with regard to group assignment.

**Gap climbing assay.** Three to five days old male flies of *Drosophila melanogaster* got their wings clipped under cold-anaesthesia and were placed back for recovery on standard cornmeal medium for 16–24 h. Then they were tested individually in the climbing paradigm on a 4 mm wide, 10 mm high and 35 mm long catwalk made of black polycarbonate with a just insurmountable 4.5 mm wide, 6 mm deep gap in the centre (Fig. 1a inset). The catwalk is situated on a circular island of 55 mm diameter which is surrounded by water and a cylindrical white shield. Ramps with a slope of 45° on either end guide the fly onto the catwalk. Ten approaches to the gap were evaluated per animal (for details, see ref. 7). The fraction of climbing attempts, recognizable by a typical 'leg-over-head behaviour', were registered (Pretest). After 3–4 days of either vibrational stress or control treatment the flies were retested in the climbing paradigm under the same conditions as in the pretest (Test 1). Depending on the experiment, the same flies were then put on pharmacological solutions overnight (14 h) and retested again individually on the next day after another either stress or control treatment (Test 2). All climbing tests were performed within the same early afternoon time window to prevent differences due to circadian activity changes. In all experiments involving vibration treatment flies of all groups were preselected to have shown four or more climbing attempts in the pretest.

**Walking activity.** Spontaneous walking activity was measured in Buridan's paradigm (for details, see ref. 12). Male flies 3-to-5-days of age got their wings clipped as described above and were subjected to either stress or control treatment for 3–4 days. Afterwards, they were individually introduced to the Buridan's arena. Two dark vertical stripes of 11° horizontal and 54° vertical viewing angle were shown on opposite sides of a translucent Perspex cylinder of 20 cm diameter illuminated by four tungsten ring lights from behind. Flies walked on an 8.5 cm diameter, water-surrounded platform located in the centre of the cylinder (29 °C, 3,000 cd m$^{-2}$) and were video tracked. Their walking activity was calculated as the total time spent walking divided by the 15 min observation time per fly in %.

**Courtship suppression assay.** Sexually inexperienced male flies were collected. For the courtship suppression conditioning they were individually grouped with one mated and therefore unreceptive female in a courtship chamber over 30 min twice a day; (for details, see ref. 13). After every session in the courtship chamber, the male fly was put back on standard cornmeal medium. The mated females were exchanged after every session. The treatment continued over 3–4 consecutive days. After the last training session in the courtship chamber either the male's courtship latency time with a new mated female or its rate of climbing initiation was determined. Courtship latency here is the time from inserting the male fly into the courtship chamber until the first wing vibration indicating courtship song.

**Fast phototaxis behaviour.** Benzer's countercurrent apparatus was used as described in ref. 14. In brief, groups of 20–50 male flies with intact wings, either after vibrational stress or control treatment, were transferred to the first of a row of six test-tube-like vials (14 mm diameter, 70 mm length). Their open sides faced the open sides of a row of five vials, which is relocatable by one position. The first vials were matched and flies shaken down to the bottom of the starting vial. Swiftly, the

apparatus was placed horizontally on a black table with light coming from the far end (Tungsten light Philips 20 W/33) in an otherwise dark room. Flies started running towards the light. After 6 s the relocatable row was shifted by one position, all flies were shaken down into the stationary row of vials and the relocatable row was shifted back. This procedure separates fast flies, which are now in vial 2, from slow flies, which remain in the starting vial; it was repeated five times per group of flies. The short time allowance of 6 s renders the paradigm sensitive for walking speed. The performance index indicates the median percentage of transitions to the light. After five separation steps 0% of all possible transitions is assigned to each fly in the starting vial, 20% to each fly in vial 2, and so on to 100% to each fly in vial 6. According to Benzer there is no stampede effect so that the flies take individual decisions to stay or to run to the light. The number of repetitions per genotype therefore equals the number of flies tested.

**Stop-for-sweet paradigm.** Male flies, with their wings cut were pretested for climbing, stress treated, and tested in climbing for their depressive-like state on the third day. On day 4 they received 6 h of vibration (without food) and were then individually tested in the stop-for sweet paradigm. Two parallel rectangular chambers of $55 \times 20\,mm^2$ were cut out of a 3 mm thick white foam board (inner margin 14 mm). The outside of the board was cut to fit in the lid of a 96-well plate ($125 \times 82\,mm^2$). A fly with cut wings was placed in each chamber and the chambers covered with a cut-to-size filter paper (filter paper grade 595½, medium fast, thin, Ø 185 mm, Schleicher & Schuell). A 5-mm-wide trace of glycerol (99.5% p.a. Roth) had been painted beforehand along the midline of the paper with a fine paintbrush. The bottom of the 96-well plate was then placed on top of the paper. Now the two flies were shaken down to the bottom of their chambers and the well plate held over vertical (110°–120°) which made the flies walk on the paper rather than on the lid. For each fly it was scored whether it overrun the glycerol or stopped to eat. Shakedown was repeated ten times before the next flies were introduced.

**Pharmacology.** All drugs were fed at concentrations shown to be effective in *Drosophila*; Lithium chloride (LiCl; 5 mM or 50 mM)[19,20], 5-hydroxy-L-tryptophan (1 mg ml$^{-1}$; Sigma H9772)[31], α-methyl-DL-tryptophan (20 mM; Sigma M8377)[48], 8-OH-DPAT (3 mM; Tocris 0529)[39] and Fluoxetine Hydrochloride (10 mM; Sigma PHR-1394)[19,35] were fed to the flies solved either in blue stained deionized water (Patent blue V sodium salt; Sigma 21605) or in blue stained 5% sucrose/arabinose (Sigma A3131) solution on filter paper for 14 h overnight. Blue staining of their abdomen served as evidence that the solution was ingested.

**Immunohistochemistry.** Adult whole-mounts were immuno-labelled following a standard protocol for fly brains as described in Thum et al.[49]. For anti-serotonin immunohistochemistry brains were dissected in cold PBS, fixed with 4% paraformaldehyde/PBS for 1 h on ice and for an additional hour at 25 °C. All subsequent washings were performed with PBTx-0.3% (PBS 0.3% Triton-X100). Brains were permeabilized for 1 h in PBTx-1% before blocking. Blocking (>1 h) and antibody incubations were performed in PBTx-0.3% with 5% normal horse serum (Vector Laboratories, S-2000). Incubation with anti-GFP (1:2,000; chicken; Aves Labs, GFP-1020), anti-mCherry (1:200; goat; Acris AB0040), anti-serotonin (1:1,000; rabbit; Sigma S5545), anti-Trio (1:100; mouse; DSHB clone #9,4A) or anti-Fasciclin II (1:100; mouse; DSHB clone #1D4) was carried out for 48 h at 4 °C. Secondary antibodies anti-chicken A488 (Invitrogen #A11039), anti-mouse Cy3 Jackson (Jackson Immuno Research #115-165-003), anti-goat Cy3 (Jackson Immuno Research #705-165-147) and anti-rabbit Cy3 (Jackson Immuno Research #111-165-003) were diluted 1:1,000 and incubated at 4 °C for 48 h. The specimens were cleared and mounted in 50% glycerol/PBS for confocal microscopy using a Leica TCS SP8 microscope and LAS AF Lite (Leica) software. For quantitative expression assessments, samples were scanned with identical settings and analysed with LAS AF Lite and ImageJ. On a 22.5 μm thick layer the R3-ring neurons of the EB were circuited in the reference pattern of 189Y-GAL4>mCD8::GFP. The diameter of the circle was increased by a factor 1.5, and the circle was transferred to the confocal stack stained against 5-HT. For quantitative measurements of the GFP or 5-HT labelling at the MB a rectangular region ($z = 22.5\,\mu m$) was drawn around the α-, β-, or γ-lobes, respectively. Within a region of interest, the grey values of all pixels with a common x-/y-coordinate were summed up along the z axis (creating voxels) and the mean of all voxels was determined. Quotients 'lobe staining' divided by 'ellipsoid body staining' were calculated.

**Statistics.** Ten approaches in climbing behaviour were annotated for each fly, the percentage of climbing attempts was calculated, and the median frequency of climbing attempts was determined over all flies of a test group. Sample size was chosen such that undisturbed wild-type behaviour in the climbing paradigm and wild-type behaviour after 3 days of stress treatment showed a highly significant difference. This sample size was kept for all genotypes and treatments. In all figures, box-whisker plots show the median (black square), 25 and 75% quartiles (box), 10 and 90% quantiles (whiskers). Since most data sets were not normally distributed, we used the Kruskal–Wallis analysis of variance with a *post-hoc* Bonferroni correction for multiple comparisons. For independent pairwise comparisons the Mann–Whitney *U* test has been applied. Statistical analyses were performed with

STATISTICA 8.0 (α level 0.05 in all cases). All statistical data are available in Supplementary Tables 1–10.

**Data availability.** The authors declare that all data supporting the findings of this study are available within this article, its Supplementary Information files or are available from the corresponding author on reasonable request.

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

## Acknowledgements

We would like to thank all colleagues who provided fly stocks that aided this study. Additional strains were obtained from the Bloomington Drosophila Stock Center (NIH P40OD018537). We thank Christina Gölz, Franziska Rieche, Olga Tschesnokowa and Tim Verlage for technical assistance and D. Kretzschmar for critical comments on the manuscript. This work was supported by a PhD scholarship of the Stipendienstiftung Rheinland-Pfalz to A. Ries.

## Author contributions

B.P. and R.S. designed the study and wrote the manuscript. A.-S.R. performed all experiments (except the stop-for-sweet measurements performed by T.H.) and all statistical analyses and prepared the figures. B.P. and R.S. supervised the project.

## Additional information

**Competing interests:** The authors declare no competing financial interests.

