## [Peer Review File · Nature Communications]

Reviewers' comments:

Reviewer #1 (Remarks to the Author):

I have reviewed the paper "Serotonin modulates a depression-like state in *Drosophila* responsive to lithium treatment." The initial observation that chronic vibrations cause a significant behavioral deficit is striking and convincing. The authors then perform experiments to test whether "motivation" is compromised. Additional controls were used to try to rule out potential effects of "frustration" on the courtship deficit. They use lithium, serotonergic agents and sucrose to rescue some of the deficits. Experiments using combinations of drugs and NaChBac expression implicate the alpha lobes of the MBs and suggest that sucrose acts through 5HT pathways. Using RNAi directed against 5HT receptors they show that disruption of the fly 5HT-1A or 1B cause a complex set of effects including the "susceptibility to stress" and "sugar relief". Finally, they use NaChBac to stimulate cells expressing 1A or 1B and find that this causes increased resilience or increased susceptibility to induction of the "depression like state," the latter indicated by a change in the time course of the effects.

This is a fascinating and elegant behavioral study and will add to our understanding of serotonin's role in behavioral regulation in the fly. A major problem with the manuscript is whether or not this represents model of depression versus another behavioral state that involves the MBs can be mitigated by serotonergic agents. To some extent, this is a moot point, since any insight into the mechanisms by which serotonin can cause a change in the overall state of an animal is important. Thus, regardless of whether we call this depression, understanding the mechanisms by which serotonergic pathways return the animal to baseline will be valuable and the tools available in the fly will make this possible.

Conversely, similar to the majority of previous papers on the relationship of 5HT to depression in rodents and humans, the interpretation of the data and links to the effects of antidepressants are somewhat exaggerated making it difficult to appreciate the more potential importance of the model in our fundamental understanding of neuromodulatory signaling. Moreover, despite some controls to rule out effects other than "depression," the possibility that other physiological pathways might be altered are not really addressed. Perhaps the most obvious mitigating factor is the state of hydration and nutrition in the stressed flies. The effects of the primary stress (shaking) on disrupting these and perhaps other basic physiological processes needs to be addressed. It seems possible that poor health could be misinterpreted as a lack of motivation. The possibility that drugs and genetic interventions might affect these processes either directly or indirectly also needs to be addressed.

Another plausible explanation for the observed behavioral effects is that some, if not all, of the sensory systems of the fly were disrupted by chronic shaking. Alternatively, effects on sensation might be via disruption of MB pathways, responsible for interpretation of sensory information. It is difficult to understand why "depression" rather than alternative explanations such as this might account for the observed effects.

The authors might also consider the possibility that serotonergic manipulation of the MBs somehow reversed effects that were caused elsewhere in the brain. This would be difficult to assess but is relevant to one of underlying tenets of this paper and a large number of other papers on depression in mammals. The fact that serotonergic agents can relieve depression does not mean that problems with serotonergic pathways cause depression. Conflating these ideas has likely contributed to the lack of progress in understanding depression and its treatment. If people are to begin studying the effects of serotonergic pathways on "depression like" states or any other altered states in the fly, it would be useful to avoid the same mistake.

These criticisms do not invalidate the manuscript as a very interesting behavioral study. The model that the authors present might be used for a variety of further experiments to determine the mechanisms by which manipulation of 5HT signaling can exacerbate or reduce the effects of chronic shaking. Indeed, it is possible that reversal of these effects might be directly related to the mechanism by which serotonergic agents relieve depression in humans, adding to the potential significance of the manuscript. However, labeling the observed effects as "depression" and suggesting that subsequent treatments relieve depression seems premature. Indeed, it is possible that the impact of the paper would be greater if the authors underplayed the interpretation of the precise internal state of the fly, since readers would not be forced into deciding whether or not the flies are depressed, and accept the results as a new model for exploring the poorly understood function of serotonergic signaling in *Drosophila*.

Additional comments

"To assess if this chronic stress induces a general lack of motivation, we measured spontaneous 15min-walking activity in the Buridan paradigm and the motivation to court a female"

The effects of 5HT on visual behavior poorly understood in *Drosophila*, but there are a few clues suggesting that it does have a role in the fly and in larger insects. It seems possible that visual circuits rather than motivation could have been affected by the treatment.

Similarly, it seems difficult to discount the possibility that 5HT had a more direct effect on courtship rather than on general motivation.

"this state does not seem to have a learning component because learning mutants, deficient in cAMP-synthesis (*rutabaga1*) or -metabolism (*dunce1*)⁴, were similarly susceptible to stress (Extended Data Fig. 2)."

Is it really clear that all types of learning in fly require cAMP/PKA?

"Indeed, LiCl-treated flies were relieved from depression, suggesting that evolutionary conserved biochemical pathways are 64 involved in MDD and in stressed flies (Fig. 1f)."
Lithium also has a variety of other effects in humans.

"Reduced levels of 5-HT or its precursor L-Tryptophan have been described in MDD patients." This is highly controversial.

"...and dietary-induced depletion of L-Tryptophan can induce symptoms of depression." Only in the context of treatment with an SSRI. It does not occur in controls or even patients treated with other types of antidepressants.

The fact that sucrose has an effect might be seen as an indication that the effect was due to general physiological dysfunction rather than depression regardless of whether or not 5HT was involved. Both the gut and the Malpighian tubules are thought to be regulated by 5HT.

Reviewer #2 (Remarks to the Author):

The authors showed that repeated mechanical vibrations reduced 5-HT level in the a-lobe of the mushroom body (MB) and demotivated flies for the attempt of gap crossing. This reduced climbing attempt could be ameliorated by feeding sucrose, which is mediated by serotonin on the MB. While some data are really interesting, they are rather incomplete to understand the underlying molecular and cellular mechanisms. Below I list the points which need to be revised upon next submission.

1. While 'anti-depressant effect' of the lithium chloride treatment seems to be very important (cf. title), there is no description about the way it works or is used in MDD (Major Depressive Disorder). The authors should explain how it modifies the behavior.
2. The 'MDD' ameliorating effect of sucrose is interesting but a bit anecdotal. Are there any reports showing that sucrose feeding can help for depression? Why do the authors think the normal food, which may contain some sugar, does not have the same effect? What are the ingredients of the food in this experiment? Do other sugars work in the same way?
3. To establish this paradigm as a model of MDD, it is important to know how SSRI affects this behavior.
4. Opposing roles of different MB lobes (Figs. 3 and 4) are not well characterized. To claim "a push-pull modulation of... of the a- and g- lobes (L117)", knocking down the 5-HT_{1A} and 1B receptors with lobe-specific drivers should be performed. Inferring from the expression patterns of the inserted GAL4 drivers (Gnerer et al 2015) is not enough. What is the evidence for the specific expression of 5-HT_{1A}-GAL4 and 5-HT_{1B}-GAL4 in a/b and g Kenyon cells? If this argument is not well substantiated, the authors should tone down and focus on the differential role of 5-HT_{1A} and 1B in the MB.
5. (Fig. 2.) Correlation between depression-like state and reduced 5-HT levels in the a-lobe is weak. Did the authors examine if sugar or 5-HTP can rescue reduced 5-HT level in the a-lobe?
6. Considering the cellular activation/inactivation experiments (Fig. 4), the effect of 5-HT_{1A} and 1B on Kenyon cell excitability seems to be opposite; the former being excitatory to a/b-

lobes while the latter being inhibitory to the g-lobe. In the current model (Fig. 4e), however, this point is unclear.

7. Trh-493 labels the serotonergic DPM neurons, and they are reported to project to the whole mushroom body lobes (Lee et al 2011). The claim that the authors detect the very weak, if any, GFP signal in b-lobe (L100) is therefore questionable. How about a'/b' lobes?

8. What is plotted in Fig 1d? Countercurrent apparatus should give the distribution in different tubes.

9. The inset in Fig1A is too small to see.

10. (P2. L43- 45.) The effect of attractive odor and visual cues are interesting but I don't see how they can be associated with other results.

11. (Figs. 2C and 2D.) Hard to see what the authors meant to say.

Reviewer #3 (Remarks to the Author):

Summary:

Ries et al is a very interesting characterization of a new assay that models depressive-like behavior in *Drosophila* and maps a serotonergic circuit required for this behavior. Repeated vibrational stimuli appears to affect likelihood of a fly to cross a gap, and effect that is alleviated by LiCl. Intriguingly, this treatment also affects how much serotonin is released into the mushroom body (MB) when an animal is given sucrose. Manipulating activity of MB neurons and levels of 5HT-1B receptors in MB neurons affects the ability to cross a gap. Where as manipulating levels of 5HT-1A receptors in MB neurons affects how sucrose affects the ability to cross a gap. Combined with previous work implicating different localization of 5HT1A and 1B receptors in the MB, this work thus proposes an exciting model where long-term stress affects how much serotonin signals to the mushroom body via a 5HT1B dependent pathway, and how much a reward (sucrose) can alleviate behavioral changes induced by this stress through a 5HT1A dependent pathway.

Concerns:

1. Regarding pharmacology data: Could the authors state in the text why pharmacological treatments rather than genetic manipulations were used to modify 5HT levels? Also, considering the lack of dose response curves for LiCl, 5HT, alphaMTP, could the authors justify the doses of drugs used? Could they also highlight their evidence that each of these pharmacological treatments don't affect behaviors required to perform the test response in the main text or supplemental text?

2. When discussing extended data figure two, the authors use a flawed argument. Mutants

deficient in cAMP or phosphodiesterase that show an inability to learn in most olfactory fly assays should not be used to show that depression-like state has a learning component. There are ways an animal can learn that may not include this mechanism. Please re-word to be more specific.

3. Regarding immunohistochemistry data: The authors are making a (valid) assumption that GFP levels don't change with sucrose treatment. Could they show the raw 5HT and GFP data in extended data to verify this assumption?

4. Line 45 (Extended Data Fig 1) – The driver lines used for these experiments do not rule out potentially effect of a'b' lobes or combinatorial effects of MB neurons. The authors may want to note this in the text.

5. The data in Figure 2e-h shows a difference in variability between control and treated animals (most obvious in Fed 2g,h). This difference in variation could potentially obscure effects within the EB, B-lobes and γ -lobes. Could the authors comment on this source of variation? Is there inherently more variation in serotonin signaling in naive animals? Could the authors speculate why this might be?

6. Could the authors comment on significance of the c739 data in Figure 3h? Does changing expression of 5HT1A Receptors in c739 neurons ameliorate the effect? What does this data add to the general argument of the manuscript?

7. Have the authors verified the effectiveness of the RNAi's? Do they effectively decrease expression of all transcripts or a particular transcript of each gene in the circuits of interest? Do they affect protein expression? If this control has been published previously, please cite this work.

8. The authors cannot rule out potential any effects of decreasing 5HT1A and 1B receptors in the MB throughout development and how this could potentially affect behavior. The authors should explicitly state this.

9. The authors may want to consider rewording the model in Figure 4 so that it is not overstating their results.

a. To show that sucrose is signaling reward via Trh493 neurons, the authors should show that activation of these neurons paired with cue results in preference for that cue in absence of stimulation.

b. To show a lobe-specific requirement of 5HT1A and 1B neurons, the authors need to show adult lobe specific knock-down of each receptor affects behavior. Previous publications show GAL4 specific effect expression patterns and protein traps for 5HT1A and 1B. However, unless this has been confirmed with immunohistochemistry, the authors cannot rule out that the receptors are indeed not more broadly expressed within the MB and thus, their results may be explained by the RNAis acting in non-lobe specific neurons.

10. The authors may want to re-word some less commonly used terms used such as "Classical master-slave experiment".

11. It was initially difficult to grasp the significance of differences between the vibration-stress pathway and the additional effects on sucrose signaling. The authors may want to consider re-organizing the writing and the figures to highlight how the sucrose experiments add an additional, and important, dimension to the work presented.

12. It was also difficult to understand the figures by observation (without very careful reading of the text and figure legends). Some recommendations are:

- a. Use colors other than red and green for word coloring and box plot shading (the current version is not color-blind friendly and the shades are difficult to distinguish).
- b. Make the colors in experimental bars more obvious (the two shades within the small size of figures was difficult to distinguish)
- c. In the extended data, combining the "Pre" groups into a single group in some graphs doubled sample size, which was somewhat misleading. Would it be possible to either show raw data within the bars (small dots or circles) or differentiate between the "pre" groups as in the main paper text?
- d. Some of the text on axes is mis-aligned with the data
- e. In some cases, adding a legend explaining color may be helpful.

13. Most of the behavior, pharmacology and comparative immunohistochemistry methodology was very brief and could be expanded on in the extended data in order to encourage reproducibility.

Reviewer #1 (Remarks to the Author):

I have reviewed the paper "Serotonin modulates a depression-like state in *Drosophila* responsive to lithium treatment." The initial observation that chronic vibrations cause a significant behavioral deficit is striking and convincing. The authors then perform experiments to test whether "motivation" is compromised. Additional controls were used to try to rule out potential effects of "frustration" on the courtship deficit. They use lithium, serotonergic agents and sucrose to rescue some of the deficits. Experiments using combinations of drugs and NaChBac expression implicate the alpha lobes of the MBs and suggest that sucrose acts through 5HT pathways. Using RNAi directed against 5HT receptors they show that disruption of the fly 5HT-1A or 1B cause a complex set of effects including the "susceptibility to stress" and "sugar relief". Finally, they use NaChBac to stimulate cells expressing 1A or 1B and find that this causes increased resilience or increased susceptibility to induction of the "depression like state," the latter indicated by a change in the time course of the effects.

This is a fascinating and elegant behavioral study and will add to our understanding of serotonin's role in behavioral regulation in the fly. A major problem with the manuscript is whether or not this represents model of depression versus another behavioral state that involves the MBs can be mitigated by serotonergic agents. To some extent, this is a moot point, since any insight into the mechanisms by which serotonin can cause a change in the overall state of an animal is important. Thus, regardless of whether we call this depression, understanding the mechanisms by which serotonergic pathways return the animal to baseline will be valuable and the tools available in the fly will make this possible.

Conversely, similar to the majority of previous papers on the relationship of 5HT to depression in rodents and humans, the interpretation of the data and links to the effects of antidepressants are somewhat exaggerated making it difficult to appreciate the more potential importance of the model in our fundamental understanding of neuromodulatory signaling. Moreover, despite some controls to rule out effects other than "depression," the possibility that other physiological pathways might be altered are not really addressed. Perhaps the most obvious mitigating factor is the state of hydration and nutrition in the stressed flies. The effects of the primary stress (shaking) on disrupting these and perhaps other basic physiological processes needs to be addressed. It seems possible that poor health could be misinterpreted as a lack of motivation. The possibility that drugs and genetic interventions might affect these processes either directly or indirectly also needs to be addressed.

Another plausible explanation for the observed behavioral effects is that some, if not all, of the sensory systems of the fly were disrupted by chronic shaking. Alternatively, effects on sensation might be via disruption of MB pathways, responsible for interpretation of sensory information. It is difficult to understand why "depression" rather than alternative explanations such as this might account for the observed effects.

The authors might also consider the possibility that serotonergic manipulation of the MBs somehow reversed effects that were caused elsewhere in the brain. This would be difficult to

assess but is relevant to one of underlying tenets of this paper and a large number of other papers on depression in mammals. The fact that serotonergic agents can relieve depression does not mean that problems with serotonergic pathways cause depression. Conflating these ideas has likely contributed to the lack of progress in understanding depression and its treatment. If people are to begin studying the effects of serotonergic pathways on “depression like” states or any other altered states in the fly, it would be useful to avoid the same mistake.

In response to the concern of reviewer 1 that the applied stress affects basic physiological processes, thus overshadowing the depression-like state or inducing fatigue, we have performed additional experiments. We tested another involuntary behaviour, optomotor response to a rotating striped pattern. As in case of fast phototaxis, stressed flies performed like controls. This again rules out that the stress-treated flies were too weak or injured to execute normal walking behaviour.

To test for another hallmark of depression-like symptoms in animal models, we have developed a test for “anhedonia” and called it stop-for-sweet paradigm. Negative geotaxis is an innate safety-seeking behaviour of flies; even if they are not startled. Satiated flies will try to reach the safe ceiling of the upright-standing microtiter plate even if they walk over a sweet-tasting glycerol stripe in their path; however, hungry flies will stop for a short meal. Notably, vibration-stressed hungry flies continue their walk to the ceiling. This indicates that they lost their appetite for the sweet or they perceive it to a lesser extent. In our view “anhedonia” in *Drosophila* or other animal models could be based on either or both. At the moment we do not have enough knowledge about the stress axis in flies or the reward pathways to address this question.

Additionally, we checked if malnutrition is inducing fatigue by feeding the same concentration of arabinose instead of sucrose to stressed flies. Although, arabinose tastes similar sweet as sucrose to *Drosophila*, it does not have a caloric value for them (Fujita ref.#25). The result that flies in the depression-like state responded with a comparable relief to arabinose as to sucrose supports our interpretation that fatigue is not the cause of reduced behaviour in stressed flies; rather the sweet sensation (in anthropomorphized terms “the pleasure”) is improving the motivational state of stressed/depressed flies.

These criticisms do not invalidate the manuscript as a very interesting behavioral study. The model that the authors present might be used for a variety of further experiments to determine the mechanisms by which manipulation of 5HT signaling can exacerbate or reduce the effects of chronic shaking. Indeed, it is possible that reversal of these effects might be directly related to the mechanism by which serotonergic agents relieve depression in humans, adding to the potential significance of the manuscript. However, labeling the observed effects as “depression” and suggesting that subsequent treatments relieve depression seems premature. Indeed, it is possible that the impact of the paper would be greater if the authors underplayed the interpretation of the precise internal state of the fly, since readers would not be forced into deciding whether or not the flies are depressed, and accept the results as a new model for exploring the poorly understood function of serotonergic signaling in *Drosophila*.

We have performed additional experiments to support the pivotal role of serotonin in modulation of motivation and the depression-like state. We have confined the RNAi-mediated knock-down of the 5HT-1A and 5HT-1B receptor genes to the adult stage to exclude putative developmental defects and to minimize long-term effects to the serotonergic response in the mushroom body. In addition, we induced the respective knock-downs in sub-compartments of the mushroom body to exclude changes in 5-HT perception in other parts of the brain.

The most impressive new data came from feeding the selective serotonin reuptake inhibitor (SSRI) fluoxetine. According to the serotonin hypothesis of depression reduced 5-HT signalling in the hippocampus of vertebrates is one of the endpoints upon permanent, uncontrollable stress. In our *Drosophila* model we observed a similar effect of continuing stress at the α -lobes of the mushroom body. SSRIs increase the serotonin levels in the synaptic cleft, thus strengthening and prolonging signalling to postsynaptic partners. In our case this is the signalling of *Thr493* neurons to the 5-HT-1A receptor at the α -lobes. The treatment with the SSRI fluoxetine increased the climbing efforts of stressed flies significantly, a result that is in strong support of our hypothesis that the sugar-induced increase of overall serotonin levels in the fly brain counteracts the loss of 5-HT signalling at the α -lobes in stressed flies.

Finally, to support our push-pull hypothesis of serotonergic signalling at the mushroom body we have treated unstressed flies with the 5-HT-1B receptor agonist 8-OH-DPAT, which in contrast to vertebrates activates the *Drosophila* 1B receptor more efficiently than type 1A (ref. Yuan #19 and Saudou #34). Overnight treatment reduced climbing efforts dramatically and this was dependent on signalling of the γ -neurons of the mushroom body.

Taken together we think that the finding that the motivation to execute different voluntary behaviours, but not exogenously induced behaviours, are reduced in vibration-stressed flies and that mood stabilising agents can ameliorate this depression-like state warrants to call it that way.

Additional comments

“To assess if this chronic stress induces a general lack of motivation, we measured spontaneous 15min-walking activity in the Buridan paradigm and the motivation to court a female”

The effects of 5HT of visual behavior poorly understood in *Drosophila*, but there are a few clues suggesting that it does have a role in the fly and in larger insects. It seems possible that visual circuits rather than motivation could have been affected by the treatment. Similarly, it seems difficult to discount the possibility that 5HT had a more direct effect on courtship rather than on general motivation.

The ability of stressed flies to compensate the rotational visual flow by optomotor response indicates that these flies have no major shortcomings in vision. Courtship latency seems to be (in part) regulated by the 5-HT-7Dro receptor (Becnel *et al.* 2011 PLoS one 6(6): e20800) which is preferentially expressed in ring neurons of the ellipsoid body, where we did not find changes in 5-HT levels.

“this state does not seem to have a learning component because learning mutants, deficient in cAMP-synthesis (*rutabaga1*) or -metabolism (*dunce1*)⁴, were similarly susceptible to stress

(Extended Data Fig. 2).”

Is it really clear that all types of learning in fly require cAMP/PKA?

To the best of our knowledge (and this includes some unpublished observations) cAMP signalling is involved in all learning paradigms that were studied in *Drosophila* so far (reviewed in Kahsai and Zars ref. #9).

“Indeed, LiCl-treated flies were relieved from depression, suggesting that evolutionary conserved biochemical pathways are 64 involved in MDD and in stressed flies (Fig. 1f).”
Lithium also has a variety of other effects in humans.

“Reduced levels of 5-HT or its precursor L-Tryptophan have been described in MDD patients.”
This is highly controversial.

“...and dietary-induced depletion of L-Tryptophan can induce symptoms of depression.” Only in the context of treatment with an SSRI. It does not occur in controls or even patients treated with other types of antidepressants.

The fact that sucrose has an effect might be seen as an indication that the effect was due to general physiological dysfunction rather than depression regardless of whether or not 5HT was involved. Both the gut and the Malpighian tubules are thought to be regulated by 5HT.

In response to concerns about the described role of L-Tryptophan/5-HT-levels and the action of LiCl we have made appropriate changes to the manuscript. The concerns about fatigue have been addressed by the experiment feeding arabinose and additional behavioural tests (see above).

Reviewer #2 (Remarks to the Author):

The authors showed that repeated mechanical vibrations reduced 5-HT level in the a-lobe of the mushroom body (MB) and demotivated flies for the attempt of gap crossing. This reduced climbing attempt could be ameliorated by feeding sucrose, which is mediated by serotonin on the MB. While some data are really interesting, they are rather incomplete to understand the underlying molecular and cellular mechanisms. Below I list the points which need to be revised upon next submission.

1. While 'anti-depressant effect' of the lithium chloride treatment seems to be very important (cf. title), there is no description about the way it works or is used in MDD (Major Depressive Disorder). The authors should explain how it modifies the behavior.

The biochemical effects of LiCl treatment are very diverse and far from understood. Li⁺ ions are competing with Mg²⁺, a cofactor of many enzymes. In insects and vertebrates LiCl negatively regulates the Shaggy (GSK3 β) protein kinase. In recent years the GSK3 signalling pathway entered the focus of depression research. We have added some information regarding SGG/GSK3 β to the manuscript.

2. The 'MDD' ameliorating effect of sucrose is interesting but a bit anecdotal. Are there any reports showing that sucrose feeding can help for depression? Why do the authors think the normal food, which may contain some sugar, does not have the same effect? What are the ingredients of the food in this experiment? Do other sugars work in the same way?

As we show, relief from the depression-like state in flies requires a sugar rush. Our standard procedure for stressing flies with vibrations over the day comprises a regular nocturnal return to the corn flour standard fly food, which contains 4% sugar-beet molasses (total carbohydrates 70%) and 4% barley malt syrup (>55% maltose, <12% dextrin); this is not sufficient to prevent the depression-like state. Presumably, the taste of corn flour might mask the sweetness of the sugars that are included; we have not tasted it ourselves. We cannot comment on the pleasure a fly takes from sweets nor do we recommend extensive sugar diets for human patients.

We have added a new experiment in which we fed sweet-tasting arabinose to stressed flies and observed a relief from the depression-like state. As this sugar does not have a nutritional benefit for *Drosophila* (Fujita ref.#25), this result indicates that it is the sweet sensation that mediates the relief and not the caloric value. With regard to humans, transient comforting effects of chocolate are described (short-term is what we see in flies), but in the long run it seems to prolong rather than relieve the dysphoric mood. "It is not, as some would claim, an antidepressant." (Review: G. Parker et al. (2006). *Journal of Affective Disorders* 92, 149–159). A review by L. Christensen (1997; *The effect of carbohydrates on affect. Nutrition* 13 (6) 503-514) states "The studies clearly suggest that carbohydrates, especially sweet simple carbohydrates, affect the mood of depressed individuals. It is not clear, however, whether

consumption of carbohydrates is beneficial or detrimental or if it is beneficial for certain depressed individuals and detrimental for others”.

3. To establish this paradigm as a model of MDD, it is important to know how SSRI affects this behavior.

We have added one experiment feeding the SSRI Fluoxetine overnight to stressed flies. This treatment did ameliorate the depressive-like state similarly to sucrose.

4. Opposing roles of different MB lobes (Figs. 3 and 4) are not well characterized. To claim “a push-pull modulation of... of the a- and g- lobes (L117)”, knocking down the 5-HT1A and 1B receptors with lobe-specific drivers should be performed. Inferring from the expression patterns of the inserted GAL4 drivers (Gnerer et al 2015) is not enough. What is the evidence for the specific expression of 5-HT1A-GAL4 and 5-HT1B-GAL4 in a/b and g Kenyon cells? If this argument is not well substantiated, the authors should tone down and focus on the differential role of 5-HT1A and 1B in the MB.

We have included additional knockdown experiments against the 5-HT-1A and 5-HT-1B receptor genes using the α -/ β -lobe or γ -lobe specific driver lines c739 or H24, respectively. We have also included the expression patterns of the 5-HT-1A- and 5-HT-1B-GAL4 driver lines to the supplementary data.

5. (Fig. 2.) Correlation between depression-like state and reduced 5-HT levels in the a-lobe is weak. Did the authors examine if sugar or 5-HTP can rescue reduced 5-HT level in the a-lobe?

Our own data (Fig. 3c and Supplementary Fig. 3) and those of Lee *et al.* (ref. #27) show that sucrose induces elevated levels throughout the fly’s brain. We therefore have refrained from repeating the 5-HT measurements of stressed flies after sucrose supplementation.

6. Considering the cellular activation/inactivation experiments (Fig. 4), the effect of 5-HT1A and 1B on Kenyon cell excitability seems to be opposite; the former being excitatory to a/b-lobes while the latter being inhibitory to the g-lobe. In the current model (Fig. 4e), however, this point is unclear.

We have added a new experiment supporting our push-pull model of behavioural motivation by the different mushroom-body compartments. The 5-HT-1B agonist 8-OH-DPAT induces a strong reduction of climbing attempts in unstressed flies. This pharmacological effect is mediated through the γ -neurons of the mushroom body, because it could be cancelled out by the expression of TNT in these Kenyon cells.

7. Trh-493 labels the serotonergic DPM neurons, and they are reported to project to the whole mushroom body lobes (Lee et al 2011). The claim that the authors detect the very weak, if any, GFP signal in b-lobe (L100) is therefore questionable. How about a'/b' lobes?

Lee et al. reported that the *Thr493*-GAL4 line addresses serotonergic neurons that innervate all compartments of the mushroom body (shown in a total z-projection in their supplement). In our hands most of the GFP signal localizes to the α - and γ -lobes. We do find significant labelling of the β' -lobes, and weak and next to no innervation of α' - and β -lobes as displayed in figure 4. We have revised the text accordingly.

8. What is plotted in Fig 1d? Countercurrent apparatus should give the distribution in different tubes.

The distribution of flies after the fifth shake-down and following run to the light is scored as follows: one of five possible transitions to the light brings flies into the first tube next to the starting tube; they have achieved 20% of all possible transitions. Flies in the second tube after the starting tube, for instance, have achieved 40% of all possible transitions to the light, and so on up to flies in the fifth tube after the starting tube, which have achieved 100% of the possible transitions. Benzer (1967) had shown that there is no stampede effect (50 flies in one experiment behaved like one fly per 50 experiments); rather every fly decides individually whether it will transit into the next vial. Therefore, we can calculate the median number of transitions to the light from 20 to 50 flies tested in one experiment. Note that Benzer had given 30s to isolate blind flies, whereas we gave only 6s to find slow-walking flies.

9. The inset in Fig1A is too small to see.

We have changed the overall size of the figures and the lettering to increase the readability.

10. (P2. L43- 45.) The effect of attractive odor and visual cues are interesting but I don't see how they can be associated with other results.

The experiments on the modulation of climbing activity through internal state and external cues are necessary to establish climbing attempts as a valid behaviour to measure motivation. These data support the notion that the MBs acts as a "limbic system" of the flies and integrate more sensory modalities than just odours. Moreover, that TNT-mediated silencing of the α -lobes impedes the influence of sensory cues on motivation, was a starting point to assess the 5-HT signalling to the sub-compartments of the mushroom bodies.

11. (Figs. 2C and 2D.) Hard to see what the authors meant to say.

We have revised the figure and added (upon request) another statistical evaluation to the supplement concerning the expression levels of GFP after stress.

Reviewer #3 (Remarks to the Author):

Summary:

Ries et al is a very interesting characterization of a new assay that models depressive-like behavior in *Drosophila* and maps a serotonergic circuit required for this behavior. Repeated vibrational stimuli appears to affect likelihood of a fly to cross a gap, and effect that is alleviated by LiCl. Intriguingly, this treatment also affects how much serotonin is released into the mushroom body (MB) when an animal is given sucrose. Manipulating activity of MB neurons and levels of 5HT-1B receptors in MB neurons affects the ability to cross a gap. Where as manipulating levels of 5HT-1A receptors in MB neurons affects how sucrose affects the ability to cross a gap. Combined with previous work implicating different localization of 5HT1A and 1B receptors in the MB, this work thus proposes an exciting model where long-term stress affects how much serotonin signals to the mushroom body via a 5HT1B dependent pathway, and how much a reward (sucrose) can alleviate behavioral changes induced by this stress through a 5HT1A dependent pathway.

Concerns:

1. Regarding pharmacology data: Could the authors state in the text why pharmacological treatments rather than genetic manipulations were used to modify 5HT levels? Also, considering the lack of dose response curves for LiCl, 5HT, alphaMTP, could the authors justify the doses of drugs used? Could they also highlight their evidence that each of these pharmacological treatments don't affect behaviors required to perform the test response in the main text or supplemental text?

To establish a useful animal model of depressive disorders is has to be amenable to pharmacological treatment therefore we used this approach in addition to transgenic manipulations. All concentrations used for pharmacological manipulations are based on published data for effective doses in *Drosophila*. We have added this information and the relevant citations to the manuscript.

2. When discussing extended data figure two, the authors use a flawed argument. Mutants deficient in cAMP or phosphodiesterase that show an inability to learn in most olfactory fly assays should not be used to show that depression-like state has a learning component. There are ways an animal can learn that may not include this mechanism. Please re-word to be more specific.

To the best of our knowledge (and this includes some unpublished observations) cAMP signalling is involved in all learning paradigms that were studied in *Drosophila* so far (reviewed in Kahsai and Zars ref. #9). We have to admit, however, that some aspects of the depressive-like state might involve learning independent of cAMP levels and have changed the text accordingly.

3. Regarding immunohistochemistry data: The authors are making a (valid) assumption that GFP

levels don't change with sucrose treatment. Could they show the raw 5HT and GFP data in extended data to verify this assumption?

We have made comparative GFP expression measurements and could not find a difference. The statistical analysis of GFP and 5-HT is included in the supplement.

4. Line 45 (Extended Data Fig 1) – The driver lines used for these experiments do not rule out potentially effect of a'b' lobes or combinatorial effects of MB neurons. The authors may want to note this in the text.

The driver lines used to assess the role of the mushroom body in climbing motivation are supposed to address all compartments including the α' - β' -neurons (c305 and NP1131; ref. #30 Aso). Of course, we cannot exclude that some Keynon cells are spared when using these lines and that there are minor contributions by the α' - β' -neurons to the motivational state of the flies. We have tuned our statement down in the manuscript.

5. The data in Figure 2e-h shows a difference in variability between control and treated animals (most obvious in Fed 2g,h). This difference in variation could potentially obscure effects within the EB, B-lobes and γ -lobes. Could the authors comment on this source of variation? Is there inherently more variation in serotonin signaling in naive animals? Could the authors speculate why this might be?

We assume that the higher variability in 5-HT levels in the control flies is due to the social stress that they nevertheless experience during the containment in the tubes. This mild stress might affect some of the control flies, whereas the vibrated flies experience massive stress and therefore a more defined reduction in 5-HT levels.

6. Could the authors comment on significance of the c739 data in Figure 3h? Does changing expression of 5HT1A Receptors in c739 neurons ameliorate the effect? What does this data add to the general argument of the manuscript?

Figure 3h (now Fig. 4h) and the new Figure 5c on c739-GAL4 driven RNAi against the 5-HT1A receptor gene supports that sugar relief is mediated via 5-HT1A receptors and α/β -neurons.

7. Have the authors verified the effectiveness of the RNAi's? Do they effectively decrease expression of all transcripts or a particular transcript of each gene in the circuits of interest? Do they affect protein expression? If this control has been published previously, please cite this work.

The 5HT-1A knock-down line #25834 has been validated by RT-qPCR in Silva *et al.* 2014 (ref#34) and Williams *et al.* 2014 (ref.#37). The 5HT-1B knock-down line #27634 has been validated by Western blot analysis in Yuan *et al.* 2005 (ref.#19). To exclude off-target effects we used additional lines #33885 and #54006 in the experiments in figure 5c-d.

8. The authors cannot rule out potential any effects of decreasing 5HT1A and 1B receptors in the MB throughout development and how this could potentially affect behavior. The authors should explicitly state this.

We have added conditional knock downs of 5HT-1A and -1B receptors to the supplement.

9. The authors may want to consider rewording the model in Figure 4 so that it is not overstating their results.

a. To show that sucrose is signaling reward via *Trh493* neurons, the authors should show that activation of these neurons paired with cue results in preference for that cue in absence of stimulation.

A permanent increase of the neuronal activity in the serotonergic *Trh493* neurons (NaChBac data shown in Fig.4b) did not change the spontaneous climbing efforts. This suggests, that the cue (i.e. the gap) is not represented through the activity of the *Trh493* neurons.

Acute activation of these neurons was not possible because the existing optogenetic transgenes (e.g. UAS-ReaChR) induced excessive climbing even without a GAL4 driver or they require too much light to be activated, which would affect vision in the climbing paradigm. Similarly, temperature-controlled induction (UAS-TrpA1) of *Trh493* neurons was not possible, because leak expression from the transgene also induced excessive climbing.

Moreover, we consider signalling of the *Trh493* neurons not as a “reward” in the same way as it is seen in associative memory formation. If flies learn something while they are vibrated, it bears no meaning when confronted with the insurmountable gap.

b. To show a lobe-specific requirement of 5HT1A and 1B neurons, the authors need to show adult lobe specific knock-down of each receptor affects behavior. Previous publications show GAL4 specific effect expression patterns and protein traps for 5HT1A and 1B. However, unless this has been confirmed with immunohistochemistry, the authors cannot rule out that the receptors are indeed not more broadly expressed within the MB and thus, their results may be explained by the RNAis acting in non-lobe specific neurons.

We have included additional knock-down experiments against the 5-HT-1A and -1B receptor genes using the α/β -lobe and γ -lobe specific driver lines *c739-GAL4* and *H24-GAL4*, respectively. We have also included the expression patterns of the 5-HT-1A- and 5-HT-1B-GAL4 driver lines to the supplementary data.

10. The authors may want to re-word some less commonly used terms used such as “Classical

master-slave experiment”.

Thank you; we have rewritten this part of the manuscript.

11. It was initially difficult to grasp the significance of differences between the vibration-stress pathway and the additional effects on sucrose signaling. The authors may want to consider re-organizing the writing and the figures to highlight how the sucrose experiments add an additional, and important, dimension to the work presented.

We have added a new experiment feeding the sweet tasting arabinose to stressed flies and observed a similar relief from frustration, as observed with sucrose. As arabinose does not have a nutritional benefit for *Drosophila*, this result indicates that it is the sweet sensation that mediates the relief and not the caloric value. We have added this interpretation to the manuscript.

12. It was also difficult to understand the figures by observation (without very careful reading of the text and figure legends). Some recommendations are:

- a. Use colors other than red and green for word coloring and box plot shading (the current version is not color-blind friendly and the shades are difficult to distinguish).
- b. Make the colors in experimental bars more obvious (the two shades within the small size of figures was difficult to distinguish)
- c. In the extended data, combining the “Pre” groups into a single group in some graphs doubled sample size, which was somewhat misleading. Would it be possible to either show raw data within the bars (small dots or circles) or differentiate between the “pre” groups as in the main paper text?
- d. Some of the text on axes is mis-aligned with the data
- e. In some cases, adding a legend explaining color may be helpful.

We have exchanged all green by blue colours. We use green and magenta for double labelling. Colours in box blots are nevertheless just an additional help, but one can understand each experiment solely by the x-axis labels. We changed the overall size of the figures and the lettering to increase the readability. We have extended the figure legends where necessary.

13. Most of the behavior, pharmacology and comparative immunohistochemistry methodology was very brief and could be expanded on in the extended data in order to encourage reproducibility.

The method section has been adjusted accordingly.

REVIEWERS' COMMENTS:

Reviewer #1 (Remarks to the Author):

I have read the revised version of the manuscript "Serotonin modulates a depression-like state in *Drosophila* responsive to lithium treatment". The authors have done a very thorough job in revising the text and added several interesting experiments in response to the previous reviews.

I have a few remaining, small suggestions.

1) Use a word other than "master" for the master-yoke experiment, as suggested by another reviewer.

2) Use a less colloquial and less speculative phrase than "sucrose rush". e.g. something simple like "behavioral response to sucrose".

3) In the sentence "Therefore, we tested if the mood stabilizing salt lithium chloride (LiCl), which is used for medication..." Rephrase "which is used for medication"

Reviewer #2 (Remarks to the Author):

I read the revised manuscript and responses to my comments. In my opinion, this version of the paper is ready for publication, if the following minor points are addressed.

1. (P7. 2nd paragraph) The authors state "Indeed, as reported earlier by Chiang and colleagues²⁷, feeding 5% sucrose overnight elevated 5-HT signals in the brain and adding the α -MTP inhibitor to the sugar solution impeded this effect (Fig. 4c; quantification in Supplementary Fig. 2c-d).

The citation should be wrong, as I don't see the data in the citation.

2. Despite the authors' response, I still don't see the necessity of Fig. S1.

Reviewer #3 (Remarks to the Author):

The authors have exceeded expectations in addressing stated concerns. No further major revisions are recommended. This work is an important contribution as it demonstrates that flies are useful for investigating the neurological and genetic basis underlying how stress affects motivated response (and possibly depression). However, after reading all of the reviewer's comments, I have to agree with an important point that reviewer #1 brings up, which isn't addressed in the revised manuscript. Reviewer 1 states, "the fact that serotonergic agents can relieve depression does not mean that serotonergic pathways

cause depression. Conflating these ideas has likely contributed to the lack of progress in understanding depression and its treatment. If people are to begin studying the affects of serotonergic pathways of depression-like states... in the fly, it would be useful to avoid the same mistake." The fly is a very powerful model to understand the biological basis of neurological disease, including MDD, and I think Reviewer 1's point is an important one to consider. If the authors do not want to change the tone of the manuscript, they may want to consider highlighting this point in the discussion, to set a precedent for future trainees reading this important work.

Minor Concerns:

Introduction:

Consider rephrasing "master" fly as "experimental" or "trained" fly.

Results:

Consider changing the word "frustration" in "the lack of motivation to climb after chronic stress was not a general response to frustration..." and "in contrast to the males that were sexually frustrated". You cannot objectively state that repeated sexual rejection over several days leads to "frustration" in flies since any effects induced may be due to a change in sensory pathways inducing changes in NPF, which in turn affects other behaviors such as ethanol consumption.

Discussion:

typo "fulfil" should be "fulfill"

Is reference #41 cited appropriately? Strausfeld & Hirth discuss the central complex and fan-shaped body rather than mushroom body.

Figure legends:

Typo in Figure 2 "Neuropeptid-F"

Figure 2, again consider rephrasing "frustration" (stated twice in legend)

Figure 2, Consider rephrasing "natural pattern" as "endogenous pattern" and citing the work that showed this (Lee et al, PNAS, 2006).

Reminder: page numbers are helpful for reviewers.

manuscript NCOMMS-16-18542B by Ries et al.

We thank our reviewers for the careful and affirmative assessment of our manuscript. Our answers are in blue typeset.

Reviewer #1 (Remarks to the Author):

I have read the revised version of the manuscript "Serotonin modulates a depression-like state in *Drosophila* responsive to lithium treatment". The authors have done a very thorough job in revising the text and added several interesting experiments in response to the previous reviews.

I have a few remaining, small suggestions.

1) Use a word other than "master" for the master-yoke experiment, as suggested by another reviewer.

A: We replaced "master fly" by "fly in closed-loop"

2) Use a less colloquial and less speculative phrase than "sucrose rush". e.g. something simple like "behavioral response to sucrose".

A: We replaced "A sucrose rush" by "Sucrose treatment"

3) In the sentence "Therefore, we tested if the mood stabilizing salt lithium chloride (LiCl), which is used for medication..." Rephrase "which is used for medication"

A: We replaced "which is used for medication" by "which is used as an augmenting medication for MDD in humans"

Reviewer #2 (Remarks to the Author):

I read the revised manuscript and responses to my comments. In my opinion, this version of the paper is ready for publication, if the following minor points are addressed.

1. (P7. 2nd paragraph) The authors state "Indeed, as reported earlier by Chiang and colleagues²⁷, feeding 5% sucrose overnight elevated 5-HT signals in the brain and adding the α -MTP inhibitor to the sugar solution impeded this effect (Fig. 4c; quantification in Supplementary Fig. 2c-d).

The citation should be wrong, as I don't see the data in the citation.

A: The relevant information is in the Supplement of paper 27 in Suppl. Fig. S1. We added "...by Chiang and colleagues (supporting information)²⁷,..."

2. Despite the authors' response, I still don't see the necessity of Fig. S1.

A: The depression-like state reduces activity - mediated by the mushroom bodies. In Fig.S1 we show, that the activity can also be increased beyond the normal state of a sated fly without sensory cues, again mediated by the mushroom bodies. Both pieces of information together show that MBs can drive behavioral activity in both directions and do this in a natural context. Moreover, the experiments of Fig.S1 on the modulation of climbing activity through internal state and external cues establish climbing attempts as a valid behavior to measure motivation. And the experiments were the starting point to assess the sub-compartments of the mushroom bodies for their role in activity control in both directions. We prefer to show this closely related information in the supplement.

Reviewer #3 (Remarks to the Author):

The authors have exceeded expectations in addressing stated concerns. No further major revisions are recommended. This work is an important contribution as it demonstrates that flies

are useful for investigating the neurological and genetic basis underlying how stress affects motivated response (and possibly depression). However, after reading all of the reviewer's comments, I have to agree with an important point that reviewer #1 brings up, which isn't addressed in the revised manuscript. Reviewer 1 states, "the fact that serotonergic agents can relieve depression does not mean that serotonergic pathways cause depression. Conflating these ideas has likely contributed to the lack of progress in understanding depression and its treatment. If people are to begin studying the affects of serotonergic pathways of depression-like states ... in the fly, it would be useful to avoid the same mistake." The fly is a very powerful model to understand the biological basis of neurological disease, including MDD, and I think Reviewer 1's point is an important one to consider. If the authors do not want to change the tone of the manuscript, they may want to consider highlighting this point in the discussion, to set a precedent for future trainees reading this important work.

A: We agree and added a statement in the discussion (p.12). "Future studies will have to show whether additional neurotransmitter systems like the dopaminergic system are affected in the depression-like state of flies." And a few lines further down on p.12 the following sentence was already in the discussion: "Several neurotransmitter systems have been implicated to be affected by Li action, including serotonin⁴³."

Minor Concerns:

Introduction:

Consider rephrasing "master" fly as "experimental" or "trained" fly.

A: We replaced "master fly" by "fly in closed-loop"

Results:

Consider changing the word "frustration" in "the lack of motivation to climb after chronic stress was not a general response to frustration..." and "in contrast to the males that were sexually frustrated". You cannot objectively state that repeated sexual rejection over several days leads to "frustration" in flies since any effects induced may be due to a change in sensory pathways inducing changes in NPF, which in turn affects other behaviors such as ethanol consumption.

A: We replaced "courtship frustration" by "sexual deprivation following sexual rejection by mated females" at the first occurrence in the text and "sexual frustration" by "sexual rejection and deprivation" at all later instances.

Discussion:

typo "fulfil" should be "fulfill"

A: corrected

Is reference #41 cited appropriately? Strausfeld & Hirth discuss the central complex and fan-shaped body rather than mushroom body.

A: Yes, the homology is shown in Fig.2 and the book by Nick Strausfeld "Arthropod brains" (2012) is cited in the figure legends. We decided to cite the Science article, as the book is less easily available.

Figure legends:

Typo in Figure 2 "Neuropeptid-F"

A: corrected

Figure 2, again consider rephrasing "frustration" (stated twice in legend)

A: corrected to "sexual rejection and deprivation"

Figure 2, Consider rephrasing "natural pattern" as "endogenous pattern" and citing the work that showed this (Lee et al, PNAS, 2006).

A: done and cited Lee et al as reference 50

Reminder: page numbers are helpful for reviewers.

A: included